



# Black carbon variability since preindustrial times in Eastern part of Europe reconstructed from Mt Elbrus, Caucasus ice cores

Saehee Lim[1, *], Xavier Faïn[1], Patrick Ginot[1, 2], Vladimir Mikhalenko[3], Stanislav Kutuzov[3], Jean-Daniel Paris[4], Anna Kozachek[3] and Paolo Laj[1, 2]

Univ. Grenoble-Alpes, CNRS, Institut des Géosciences de l'Environnement, Grenoble, France.
Univ. Grenoble-Alpes, CNRS, IRD, Observatoire des Sciences de l'Univers, Grenoble, France.
Institute of Geography, Russian Academy of Sciences, Moscow, Russia.
Laboratoire des Sciences du Climat et de l'Environnement, IPSL, CEA-CNRS-UVSQ, CE Orme des Merisiers, 91190 Gif sur Yvette, France.

*now at: Department of Earth and Environmental Sciences, Korea University, Seoul, South Korea.

*Corresponding to*: S. Lim (saehee.lim@gmail.com)

**Abstract** Black carbon (BC), emitted by fossil fuel combustion and biomass burning, is the second largest man-made contributor to global warming after carbon dioxide (Bond et al., 2013). However, limited information exists on its past emissions and atmospheric variability. In this study, we present the first high-resolution record of refractory BC (rBC, including mass concentration and size) reconstructed from ice cores drilled at a high-altitude Eastern European site in Mt. Elbrus (ELB), Caucasus (5115 m a.s.l.). The ELB ice core record, covering the period 1825-2013, reflects the atmospheric load of rBC particles at the ELB site transported from the European continent with a larger rBC input from sources located in the Eastern part of Europe. In the first half of the 20[th] century, European anthropogenic emissions resulted in a 1.5-fold increase in the ice core rBC mass concentrations as respect to its level in the preindustrial era (before 1850). The rBC mass concentrations increased by a 5-fold in 1960-1980, followed by a decrease until ~2000. Over the last decade, the rBC signal for summer time slightly increased. We have compared the signal with the atmospheric BC load simulated using past BC emissions (ACCMIP and MACCity inventories) and taken into account the contribution of different geographical region to rBC distribution and deposition at the ELB site. Interestingly, the observed rBC variability in the ELB ice core record since the 1960s is not in perfect agreement with the simulated atmospheric BC load. Similar features between the ice core rBC record and the best scenarios for the atmospheric BC load support that anthropogenic BC increase in the 20th century is reflected in the ELB ice core record. However, the peak in BC mass concentration observed in ~1970 in the ice core is estimated to occur a decade later from past inventories. BC emission inventories for the period 1960s-1970s may be underestimating European anthropogenic emissions. Furthermore, for summer time snow layers of the last 2000s, the slightly increasing trend of rBC deposition likely reflects recent changes in anthropogenic and biomass burning BC emissions in the Eastern part of Europe. Our study highlights that the past changes in BC emissions of Eastern Europe need to be considered in assessing on-going air quality regulation.

## 1 Introduction

Climate forcing of black carbon (BC), a primary aerosol emitted by fossil fuel and biomass combustions, is of





great concern due to its strong light-absorbing ability and small size allowing it to transport over long distances
(Bond et al., 2013; Ramanathan and Carmichael, 2008). In high-altitude or –latitude areas, BC has been
identified as a significant contributor that may accelerate snowmelt (Hansen and Nazarenko, 2004; Xu et al.,
2016). Despite numerous studies through both measurements and model simulations (Bond et al., 2013 and
references therein), little is known about BC's past variability, e.g., before year 2000, and sensitivity to climate
change primarily due to limited in-situ atmospheric BC measurements both temporally and spatially (Collaud
Coen et al., 2007, 2013).
Reconstruction of atmospheric BC variability from ice core archives can thereby be very helpful to understand
past BC emissions and provide additional constraint on BC emission inventories (Bisiaux et al., 2012a; Kaspari
et al., 2011; Legrand et al., 2007; McConnell et al., 2007; Wang et al., 2015). Particularly, the geographical
proximity of the ice cores at high-altitude Alpine sites, e.g., European Alpine sites such as Col du Dôme, Colle
Gnifetti and Fiescherhorn (Jenk et al., 2006; Legrand et al., 2013; Thevenon et al., 2009) to densely populated
regions allows us to observe a fingerprint of BC emissions and past temporal variability in the anthropogenic
source regions. In this respect, elemental carbon (EC) records reconstructed from ice cores at Western European
Alpine sites (Col du Dôme and Colle Gnifetti) highlight a pronounced EC increase starting mid-20[th] century
(Legrand et al., 2007; Thevenon et al., 2009) with increasing anthropogenic activity in the Western Europe
(Fagerli et al., 2007; Lamarque et al., 2010). However, recent EC records over the last two decades are not
available from these Western European ice cores, which makes it difficult to quantify historic BC emissions and
thus provide implications for assessing European air quality regulation initiated since 1970 (Tørseth et al., 2012;
Vestreng et al., 2007). Furthermore, long-term ice core BC records have never been reconstructed from the
Eastern European regions where even atmospheric measurements are relatively scarce (Pio et al., 2007; Yttri et
al., 2007). Reconstruction of BC records with a wide range of coverage both temporally and spatially is crucial
to understand BC emission properties and establish regulations on the emissions.
In this study, we present a high-resolution record of refractory BC (rBC) deposition to snow at a high-altitude
site in Mt. Elbrus, Caucasus (5115 m a.s.l.) covering the period 1825-2013. Located between the Black and the
Caspian seas, Mt. Elbrus is influenced by prevailing westerly from the European continent (Mikhalenko et al.,
2015). The ice core rBC record, that is reconstructed in the downstream of European continent, therefore
provides information on long-term variability and evolution of BC emissions of Europe. The study documents
the variability of rBC deposition and provides a comparison with the expected atmospheric BC variability based
on past emission inventories also considering atmospheric transport to the drilling site.

## 2. Method

### 2.1 Ice core drilling site

A 181.8 m-long ice core (the 2009 core) was drilled at the Western Plateau of Mt. Elbrus (ELB), the highest
summit of the Caucasus (43°20'53,9"N, 42°25'36,0"E, 5115 m a.s.l.) (Figure 1) on September 2009. In addition,
a 20.4 m-long ice core (the 2013 core) was extracted in June 2013 at the same site to expand the existing ice-core
sample set from 2009 to 2013. Drilling was performed in a dry borehole with a lightweight electromechanical
drilling system, and was accompanied by borehole temperature measurements. Borehole temperatures ranged
from -17 °C to 10 m depth to -2.4 °C at 181 m of the 2009 core (Mikhalenko et al., 2015).





The core were packed in polyethylene sealed bag and stored on the glacier at -10°C. After the drilling campaign,
the core were packed in insulated core boxes and shipped frozen to the cold laboratory of the Lomonosov
Moscow State University for preliminary investigation and water stables isotopes analyzes. In Moscow, the core
was split and one-half was shipped to LGGE (Laboratoire de Glaciologie et Géophysique de l'Environnement -
now Institut de Géophysique de l'Environnement) in Grenoble, France for additional analyzes.

**2.2 rBC ice core analysis**
The top 156.6 m of the 2009 core and the entire 2013 core were analyzed at LGGE in 2013-2014 and in 2014,
respectively, using an ice core melter system coupled with a jet nebulizer (APEX-Q, Elemental Scientific Inc.,
Omaha, NE) and a single particle soot photometer (SP2, Droplet Measurement Technologies, Boulder,
Colorado). We have used the terminology proposed by Petzold et al. (2013) for incandescence-based BC
measurements. Our results are therefore reported in terms of refractory-BC (rBC). It should be noted that there is
a direct relationship (although not necessarily linear) between rBC and BC measured with other techniques
(Kondo et al., 2011a; Laborde et al., 2012; Miyakawa et al., 2016).
Dust and conductivity are continuously analyzed simultaneously to rBC.  Briefly, ice core sticks (3.4 cm x 3.4
cm x 1 m) were melted at a mean rate of 3 cm min$^{-1}$ and the melt water from the inner 6.8 cm$^2$ of the sticks were
continuously collected. After de-bubbling, the sample flow is split to rBC analytical line with a mean flow of
about 70±10 µL min$^{-1}$. The flow rate dedicated to rBC analyses is continuously recorded using a mass flow meter
(SENSIRION© SLI-2000). In parallel, the melt water was sampled by two auto-samplers at the end of the CFA
for off-line ionic species analysis and archive storage. The upper section of the 2009 firn core was analyzed
discretely.
Ice core rBC analysis using the SP2 has been reported previously (Bisiaux et al., 2012a, 2012b; Ginot et al.,
2014; Jenkins et al., 2013; Kaspari et al., 2014; Wang et al., 2015). Specifically, recent papers describe detailed
analytical evaluation for rBC in liquid samples, e.g., rain, snow and ice core, using the SP2 (Lim et al., 2014;
Mori et al., 2016; Schwarz et al., 2012; Wendl et al., 2014). The SP2 uses a laser-induced incandescence method
to measure the mass of individual rBC particle (Schwarz et al., 2006; Stephens et al., 2003). Briefly, an
individual rBC particle passes through the laser beam intra-cavity of a 1,064 nm Nd YAG laser and
incandescences. Of two PMT-photo detectors (broad and narrow bands) that are used to detect incandescence
signal, we used only broadband detector to derive rBC mass avoiding low signal-to-noise ratio from the
narrowband detector. The SP2 was calibrated by analyzing mass-selected fullerene soot (Alfa Aesar Inc., USA).
The design and gain settings of our SP2 resulted in the lower and upper limit of measurements for rBC mass to
be ~0.3-220 fg. A particle larger than 220 fg was treated as the particle of 220 fg. Loss of rBC particles occurring
during aerosolization in the APEX-Q was calibrated and corrected daily by rBC standard solutions (Aquadag®,
Acheson Inc., USA; 8 steps from 0.1 to 100 µg L$^{-1}$), which resulted in rBC mass recovery of 75±7 %. The rBC
fraction that was not aerosolized was partially identified in drains and internal surface of the APEX-Q (see
Supplementary information in Lim et al. (2014)). To prevent contamination and achieve the rBC levels as low as
possible, both an instrumental blank (ultrapure water) and a 5-cm procedure blank (frozen ultrapure water cut in
the cold room) were run daily prior to field sample analysis, until the rBC counting reached 0 to 1 per second,
equivalent to rBC concentration of less than 0.01 µg L$^{-1}$.



High resolution continuous rBC data recorded every second was smoothed at a depth resolution of 1 cm, except
the upper section (surface to 7.2 m depth) of the 2009 core that was discretely analyzed at a depth resolution of
~5-10 cm. The density of rBC data points per year depends on annual snow accumulation rates and ice thinning
with depth. The two ice cores are overlap for snow layers of year 2007-2009 (Fig. S1). The records described
here for rBC concentrations are (i) the 2009 ice core from 2.9 m to 156.6 m, corresponding to calendar years of
1825-2008 and (ii) the top 15.9 m of the 2013 core, corresponding to calendar years of 2009-2013. These two ice
core records cover the calendar years of 1825-2013.
As a first survey for long-term rBC size distributions of ice core record, mass equivalent diameter of measured
single rBC, $D_{rBC}$, was calculated, assuming a void-free BC density of 1.8 g cm$^{-3}$ (Moteki and Kondo, 2010). The
calculated $D_{rBC}$ was in the range of ~70 and 620 nm. A series of test using mono-dispersed polystyrene latex
(PSL) spheres with known diameters (150-600 nm) and poly-dispersed standard BC (Aquadag®) suggests that
the APEX-Q/SP2 system preserves original size information of rBC particles in liquid samples and provides
highly reproducible rBC size measurements with a variation of < 5 nm (Sect. 2.2.3 and 2.2.5 in Lim et al., 2014;
Wendl et al., 2014). rBC size distributions were retrieved seasonally and simplified with a log-normal fit with a
bin size (#)=200. Mass mode diameter (MMD) of the log-normal fit was then extracted to further reduce
parameters. Size intervals between bin channels vary, with the minimum interval of less than 8 nm for the MMD
200-350 nm. Here, all SP2 data were processed with the SP2 toolkit developed by M. Gysel at the Paul Scherrer
Institute (PSI, Switzerland; http://aerosolsoftware.web.psi.ch/).

**2.3 Ice core dating and seasonal signature**

Ice core dating was determined by counting annual layers from 1825 to 2013 using the seasonal cycles of
ammonium, succinic acid and water stable isotopes (δD and δ$^{18}$O) that were analyzed discretely. Based on the
examination of the ammonium and succinic acid profiles, each annual layer was divided into two parts
corresponding to snow deposition under winter condition and summer condition (Legrand et al., 2013; Preunkert
et al., 2000). In addition, the annual layer counting was further confirmed using the reference horizon from a
tritium peak (1963) and a volcanic horizon (Katmai in 1912). The mean annual net accumulation rate of 1455
mm w.e. for the last 140 years was estimated from these proxies. The dating uncertainty is likely on the order of
a few years due to ice thinning with deeper depths. Further details about dating are found in Mikhalenko et al.

143   (2015).

Ice core seasonality was determined by the ammonium stratigraphy and further verified by the isotope variations.
However, seasonal separation of the high-resolution rBC record made by lower-resolution ammonium profile
was sometimes challenging particularly at the edge of two seasons, misleading winter (summer) rBC layers to be
more concentrated (less concentrated) by the adjacent seasonal rBC layer. To avoid inaccurate separation of an
annual ice layer into winter and summer intervals, only mid-summer and mid-winter rBC concentrations were
extracted by considering data comprised between the 25$^{th}$ percentile and the 75$^{th}$ percentile of the depth thickness
of each seasonal snow layer. The mid-summer and mid-winter are therefore corresponding roughly to the
warmest three months and the coldest three months ("background winter") of a year. Later in the manuscript,
summer and winter of this study will refer to mid-summer and mid-winter, respectively.





### 2.4 Atmospheric transport modeling

#### 2.4.1 Model description and runs

FLEXPART v6.2 lagrangian particle dispersion model (LPDM) calculates the trajectories of tracer particles using the mean winds interpolated from the gridded analysis field and parameterizations representing turbulence and convective transport (Forster et al., 2007; Stohl and Thomson, 1999). FLEXPART was run using reanalysis fields of the European Centre for Medium-Range Weather Forecasts (ECMWF, ERA-Interim) at 0.75°×0.75° resolution, which is available since 1979. Here, a backward simulation mode was used to analyze particles transport pathways from potential flux regions to the sampling site (Seibert and Frank, 2004; Stohl et al., 2005). To limit computational cost, simulations were performed for two selected periods: 2005-2009 and 1979-1983. We selected these periods because: (i) year 1979 is the first year of ECMWF data and year 2009 is the last year of our longer ice core (2009 ice core) that were analyzed prior to the 2013 ice core and (ii) these years are inflections in rBC trends (Sect. 3.2). It would thus be sufficient to analyze transport patterns influencing rBC at ELB and determine potential changes in these transport patterns. 1,000 particles are released at the drilling site during every 5-day interval in June to August (JJA) and in December to February (DJF). Modelled global average atmospheric lifetimes of BC particles varies by a factor of more than 3, ranging from 3 to 10 days (Bond et al., 2013). Because BC particles reaching the high-altitude ELB site would experience longer lifetimes than the particles transporting in the planetary boundary layer (PBL), simulations were performed using a BC lifetime of 5-, and 7-day. However, 7-day air mass trajectories were extending to the Pacific and therefore made little difference with the 5 days simulations. Thus we set the BC lifetime as 5-day. Number of particles were then computed every 3h at 0.5°×0.5° resolution.

#### 2.4.2 Sensitivity by potential source regions

The finally defined footprint density $F(i, j, n)$ is expressed as a parameter encompassing released particle number and residence time along the particles pathway, in procedure defined unit (p.d.u.). This final result is theoretically identical to potential emission sensitivity (PES), called source-receptor-relationship by Seibert and Frank (2004), which is proportional to the particle residence time in a particular grid cell with a fixed altitude range.

To facilitate analysis we reduced the number of variables from the gridded footprint density by summing them over large regions. We classified the footprint areas into five geographical regions with specific rBC emission sources (Figure 2). The regions identified are as follows: **EEU** (Eastern Europe including nearby the Mt. Elbrus, Ukraine and Europe Russia and a part of Middle East), **CEU** (Central Europe), **WEU** (Western Europe), **NAF** (North Africa), **NAM** (North America) **and Others** (The Atlantic and a part of Northern Europe above 60°N).

To display our results, we first calculate the footprint density $F$e of the entire footprint area:

$F\text{e}(i,j) = \sum_{n=1}^{N} F(i,j,n)$

Here, $F(i, j, n)$ is footprint density, where $i$ and $j$ are the indices of the latitude/longitude grid and $n$ runs over the total number of cases N. $F$e indicates the entire footprint area where the aerosols track during the last 5 days of transport. Note that we found little inter-annual variability in the footprint contribution of each region to the ELB site with a 3 % variation over the two periods (2005-2009 and 1979-1983). Assuming that this inter-annual





variability in footprint density is not large enough to influence on long-term rBC trends and the results over the
two periods are thus fairly representative of 20[th] century, we combined the simulations results and used this
approach to study long-term emission contribution of each geographical region to rBC distribution and
deposition at our drilling site.
In addition to the calculation using total particles in the atmospheric column, calculations using particles
positioned in the lowest 2 km layers in the atmosphere were performed to investigate emission source regions of
aerosols transporting from low altitudes. To show the potential particle transport strength of each region relative
to the entire area, we calculated the percentages of the footprint density in each region relative to the one in the
entire area.  To do this, we sum $Fe\,(i, j)$ over the entire footprint area resulting in one value. In the same way, we
sum $F\,(i, j)$ within each of the five regions resulting in five values.

**2.5 Historic BC emission inventories**
To describe temporal variability in the regional BC emissions and atmospheric load of BC transported to the
ELB site, we used time-varying anthropogenic and biomass burning BC emissions estimated by ACCMIP
(Emissions for Atmospheric Chemistry and Climate Model Intercomparison Project) inventory for the period
1900-2000 on the decadal scale (at 0. 5°×0. 5° resolution; Lamarque et al., 2010) and MACCity
(MACC/CityZEN EU projects) inventory for the year 2008 (at 0. 5°×0. 5° resolution; Diehl et al., 2012; Granier
et al., 2011; Lamarque et al., 2010; van der Werf et al., 2006). Note that the ACCMIP inventory provide decadal
means (e.g., '1980' corresponds to the mean of 1980-1989) for the biomass burning estimates and representative
values (e.g., '1980' is a representative of 1975-1985) for the anthropogenic estimates, leading to 5-year shift
between two estimates. We used anthropogenic emission only for constraining BC emissions in DJF and both
anthropogenic and biomass burning emissions for constraining BC emissions in JJA, because biomass burning
frequently occurs in summer time as respect to anthropogenic emissions occurring year-round.
**3 Results and discussion**
**3.1 High resolution rBC record from Elbrus ice cores**
We present the first high-resolution rBC record of ice cores drilled in the Mt. Elbrus, Caucasus (2009 and 2013
cores, Figure 3a). The rBC concentrations along the two cores ranged from 0.01 µg L$^{-1}$ to 222.2 µg L$^{-1}$ with a
mean±1σ of 11.0±11.3 µg L$^{-1}$ and a median of 7.2 µg L$^{-1}$. A 20-m long section is zoomed in Figure 3b to
highlight the higher resolution of rBC signals when continuously recorded at 1-cm depth interval compared to
the surface snow and firn section (from top to 6.1 m) analyzed discretely at ~5-10 cm-depth interval. The rBC
record was found to preserve sub-annual variability from top to depth of 156.6 m with rBC spikes reflecting
large and abrupt variability in deposition of atmospheric rBC particles. Such high-resolution record brings new
opportunities to study dynamic atmospheric vertical transport and/or sporadic events in a season.
A well-marked seasonal rBC cycle (e.g., Fig 3b) was characterized for the 2013 core and the 2009 core down to
156.6 m by consistent high summer values ranging from 0.2 to 222.2 µg L$^{-1}$ with a mean±1σ of 15.5±12.9 µg L$^{-1}$
and a median of 11.7 µg L$^{-1}$ and low winter values ranging from 0.2 to 44.6 µg L$^{-1}$ with a mean±1σ of 5.9±5.1
µg L$^{-1}$ and a median of 4.5 µg L$^{-1}$ (Table 1). Peak rBC mass concentration of an annual snow layer was observed



in summer snow layer. In atmospheric observations at ground-based sites in Western and Central Europe
boundary layer, EC aerosol mass concentrations in winter are higher roughly by a factor of 2 than in summer
mainly due to the enhanced domestic heating (Pio et al., 2007; Tsyro et al., 2007). In contrast to the boundary
layer sites, the atmospheric measurements at high-elevation sites in Europe (e.g., Puy de Dôme at 1465 m a.s.l.
and Sonnblick at 3106 m a.s.l.) revealed 2 to 3 times higher EC levels during summer than winter (Pio et al.,
2007; Venzac et al., 2009), reflecting the efficient upward transport of BC aerosols from the polluted boundary
layer to the high-altitudes during summer, primarily by thermally-driven convection and thickening boundary-
layer height (Lugauer et al., 1998; Matthias and Bösenberg, 2002). This is consistent to the rBC seasonality
observed in the ELB ice core.

**3.2 Long term evolution of rBC mass concentrations**
Time series of summer and winter medians of rBC mass concentrations from 1825 to 2013 are shown in Figure
4. Medians are shown with lower and upper $10^{th}$ percentiles to illustrate seasonal rBC concentrations. The rBC
concentrations varied significantly over the past ~190 years with a large inter-annual variability. Both summer
and winter rBC medians increased gradually since the onset of $20^{th}$ century with a rapid increase in ~1950 lasting
until ~1980. Median concentrations reached their maximums in the mid-1960s for summer (37.5 µg $L^{-1}$) and in
the late 1970s for winter (14.7 µg $L^{-1}$).
Concentrations and relative change to levels of preindustrial era (here, defined as 1825-1850) for given time
periods are summarized in Table 1. For the period of 1825-1850, median (± standard deviation, SD) of rBC
concentrations were 4.3±1.5 µg $L^{-1}$ in summer and 2.0±0.9 µg $L^{-1}$ in winter. The rBC concentrations increased by
a ~1.5-fold in 1900-1950. Over the period of 1960-1980, rBC concentrations increased by a factor of 5.0 in
summer and a factor of 3.3 in winter. The larger relative change of summer rBC than one of winter for the period
suggests that rBC emissions in summer source region increased more sharply for this time period. Notably, in
addition to medians, the lower $10^{th}$ percentiles of both summer and winter rBC increased since the preindustrial
era, highlighting that rBC background level in the atmosphere at ELB was also significantly modified.
Meanwhile, upper $10^{th}$ percentiles ranged up to 75 µg $L^{-1}$ and 35 µg $L^{-1}$ for summer and winter, respectively.
Of the EC records available in the Western European mountain glaciers, only Col du Dôme (hereafter, CDD;
Legrand et al., 2007) and Colle Gnifetti (hereafter, CG; Thevenon et al., 2009) summer records provide EC
records for the recent time (until ~1990 and 1980, respectively), whereas the Fiescherhorn (hereafter, FH; Jenk et
al., 2006) record is available until 1940 only. Both summer records at CDD and CG show somewhat comparable
preindustrial EC levels (~2 µg $L^{-1}$ for CDD and ~7 µg $L^{-1}$ for CG in the mid-1800s) to the ELB rBC (4.3±1.5 µg
$L^{-1}$ in 1825-1850) and substantially increased EC concentrations for the period 1950-1980 since the mid-$19^{th}$
century, similar to the ELB rBC. This suggests that EC emissions show a common trend at the European scale,
and that such trend has been recorded in the different European high-altitude ice cores from CDD, CG, and ELB.
Some differences, such as peak time period and increase/decrease rate between records that may reflect sub-
regional (e.g., Western Europe vs. Eastern Europe) emission changes, may be also noteworthy. However, direct
comparison of the ELB rBC with the Western European ice core records should be made with caution owing to
both (i) different analytical methods applied for the ice cores (e.g., ELB rBC: APEX-Q/SP2, CDD EC: thermal-
optical method with EUSAAR2 protocol, and CG EC: thermal method) and (ii) lower data resolution particularly



for the CDD core (a few data points for a decadal EC concentration). We thus focus on evaluating the ELB rBC
record in Sect. 3.5 by comparing with simulated atmospheric load of BC particles that were transported from
source regions to the Mt. Elbrus.

**3.3 Past variability in rBC size distributions**

The first record of temporal and seasonal changes in rBC size distribution was extracted from the ELB ice core.
Mass equivalent diameter of rBC particles ($D_{rBC}$) was log-normally distributed. The mode of rBC mass size
distributions (mass mode diameter, MMD) was determined for both summer and winter layers by fitting a log-
normal curve to the measured distribution (e.g., Figure S2). This approach provides reliable results of
representative rBC size in seasonal ice layers as the determined MMDs fall into the measured size range (~70-
620 nm).
Figure 5 shows time series of rBC MMD for the period of 1940 to 2009. The upper and lower limits of the
periods selected for retrieving rBC MMD were chosen so as a large number of rBC particles in the seasonal ice
layer would be available and would allow to secure reliable size distribution of the ice layer. Faster melting of
snow layers of year 2010-2013 and thinner ice layers below the layer of year 1940 did not allow to record
sufficient numbers of rBC particles and thus robust rBC size distributions could not be retrieve. For the
considered time period, rBC MMD of both summer and winter layers varied ranging from 207.3 nm to 378.3 nm
with a geometric mean of 279.4±1.1 nm. No clear temporal change in rBC MMD was identified over the 1940-
2009 period.
Notably, rBC particles measured in this study show the MMD shifted to larger sizes than those measured in the
atmosphere over Europe (MMD of 130-260 nm) (Dahlkötter et al., 2014; Laborde et al., 2013; Liu et al., 2010;
McMeeking et al., 2010; Reddington et al., 2013), even larger than atmospheric rBC diameter measured at an
high alpine site, Jungfraujoch (JFJ) in Switzerland (MMD of 220-240 nm) (Liu et al., 2010). The shift of rBC
sizes induced by dry deposition should be negligible, as wet deposition with fairly constant precipitation
throughout the year (e.g., 52% in summer and 48% in winter of annual mean precipitation at Pereval
Klukhorskiy observatory located at 2037 m a.s.l. in the Western Caucasus) is the dominant aerosol removal
pathway at this site (Mikhalenko et al., 2015). Similarly, significant snow melt was not observed in the ELB
summer ice layers and post-deposition processes are thus not expected to alter rBC size distributions. Rather, the
different rBC size distributions of the ice core from those in the atmosphere are likely associated with removal
process of rBC particles during precipitation. Recent study using the SP2 technique showed the rBC size
distribution in rainwater shifted to larger sizes (MMD= ~200 nm) than that in air (MMD= ~150 nm) in Tokyo,
indicating that large rBC particles were more efficiently removed by precipitation (Mori et al., 2016). The
preferential wet removal of larger rBC particles (Mori et al., 2016; Moteki et al., 2012) could reasonably explain
the larger MMD of rBC particles observed in the ice core than atmospheric rBC aerosols (Schwarz et al., 2013).
The seasonal variations in rBC size distribution are clearly visible. In summer, the MMD varied ranging from
227.4 nm to 378.3 nm with a geometric mean of 290.8±1.1 nm (Fig.5, red curve). In winter, the MMD varied
ranging from 207.3 nm to 344.9 nm with a geometric mean of 268.7±1.1 nm (Fig.5, blue curve). The rBC MMD
of summer ice layers tended to be slightly larger than that of winter layers. Despite few observational evidences,
we hypothesize that larger rBC size in summer may reflect advection of rBC aerosols transported from the PBL



by thermally-driven convection, while in winter aerosols transported in the free troposphere (FT) could be
smaller due to longer residence time in the atmosphere and accordingly, more chances for larger aerosols to be
removed by precipitation prior to reaching the ELB site. Our hypothesis seems to be reasonable being consistent
to the findings of in-situ aerosol measurements at high-altitude sites in Europe. Liu et al. (2010) found that rBC
aerosols at JFJ were slightly larger when the site was influenced by valley sources, anthropogenic pollutants
from lower altitudes. Submicron aerosol size distributions were also overall shifted to larger size in summer (50
to 150 nm) than in winter (below 50 nm) at European mountain stations with altitude of ~1000-3000 m a.s.l.
(Asmi et al., 2011). The authors in the latter explained this feature by relatively polluted air masses from the PBL
during daytime in summer, but more influence of the FT air masses in winter. Similar to the clear seasonal cycle
in rBC mass concentration, the clear seasonal rBC size distributions of the ELB ice core point out seasonal
differences in origins of air masses reaching the ELB drilling site: PBL air with less chance of aerosol wet
removal in summer and the free tropospheric air in winter.
In addition, the larger rBC MMD in summer layers can be associated with specific summer sources of
atmospheric rBC particles, such as forest fires and/or agricultural fires. Particularly, forest fires in Southern
Europe and agricultural fires in Eastern Europe may well contribute to summer aerosol loading in Europe
(Bovchaliuk et al., 2013; van der Werf et al., 2010; Yoon et al., 2011). Previous SP2 studies have reported the
larger size of rBC aerosols for biomass burning plumes, e.g., MMD of ~200 nm (Kondo et al., 2011b; Schwarz et
al., 2008; Taylor et al., 2014) compared to rBC sizes for urban plumes. In the ELB ice core, we observed a
maximum rBC MMD of 378.3 nm, with a maximum rBC mass concentration of 222.2 $\mu$g L$^{-1}$ in the late summer
snow layer of year 2003, when extreme forest fire events occurred over the Iberian Peninsula and the
Mediterranean coast (Barbosa et al., 2004; Hodzic et al., 2006) resulting from a record-breaking heatwave in
Europe (Luterbacher et al., 2004; Schär et al., 2004). Both forward and backward air mass trajectories calculated
from HYSPLIT model support that the ELB site was potentially influenced by the intense forest fires occurred in
the Southern part of Europe on the mid-August 2003 (Fig. S3), when the top altitude of the PBL was estimated to
be ~4.5 km high (Hodzic et al., 2006). Although speculative, this snow layer of year 2003 peaked with rBC
concentration and enriched with larger-sizes rBC particles indicates potential contribution of biomass burning
aerosols transported westerly to the ELB site.
The rBC size distributions preserved in Elbrus cores could be discussed as an influence of seasonal vertical
transport versus emission sources of rBC aerosols and their wet removal properties. This rBC size information is
potential to provide important implications particularly for the determination of snow-melting potential by rBC
particles in snow (Flanner et al., 2007; Schwarz et al., 2013). Comparison of rBC size with well-established
biomass burning proxies would be required to better characterize the dependency of rBC sizes with past fire
activities.

### 341  3.4 Potential emission source regions

Figure 6 illustrates potential source regions of BC aerosols reaching the ELB site. The model results show that
relative to the footprints in JJA, footprints in DJF were more spread out of European continent and extended
further over the Pacific (Figure 6a and b). The relative contributions of each regional footprint density over the
total density are summarized in Fig. 7. Most of aerosols reaching the ELB site are transported from the European



continent (WEU+CEU+EEU) accounting for 71.0 % and 55.6 % in JJA and DJF, respectively. The region EEU
brings the greatest contribution with fairly consistent features for both seasons, accounting for 35.6 % and
30.9 % in JJA and DJF, respectively. A stronger seasonality was found in the region NAF and the region NAM,
where the footprint contribution was larger in DJF by a 2-fold. This seasonal variation is caused by longer
particle trajectories promoted by a faster zonal flow in winter across the North Atlantic from west to east.
To investigate contributions of aerosols transporting from low altitudes which may reflect emissions at surface
more sensitively, we calculated the footprint density of particles positioned in the lowest 2 km layers in the
atmosphere. Note that we arbitrarily selected this vertical height of atmosphere (2 km layer) since particles
positioned at lower atmosphere (e.g., ~1 km layer) was rarely observed in our simulations and the PBL heights
were often higher in European mountains up to 3 km (Matthias, 2004). The results for JJA show that unlike in
the entire atmospheric column, the contribution of footprint density from the region EEU was almost doubled in
the 2 km layer, accounting for 63.6 % (Fig. 6c). Contrarily, in DJF, the proportion of the region EEU was only
22 % over total footprint density in this fixed layer. We thus infer that large seasonal increases observed during
summer time in rBC mass concentration are likely driven by deposition of rBC aerosols transported from Eastern
part of Europe and mostly originating from lower altitudes.
Therefore, these FLEXPART results confirm that rBC deposition to the Mt. Elbrus is most likely dominated by
transport of BC emissions from the European continent, with the strongest BC inputs from the Eastern part of
Europe particularly in summer.

## 3.5 New constrains on European BC emissions

Refractory BC concentrations of the ELB ice core increased rapidly from the 1950s to the 1980s (Figure 4 in
Sect. 3.2), and such trend record should primarily reflect changes in European BC emissions (Sect. 3.4). Here,
we compare past emission BC inventories with the ELB ice core record to bring new constrains on past
European BC emissions.
Figure 8 shows temporal changes in anthropogenic and biomass burning BC emissions for the period 1900-2008
estimated by ACCMIP and MACCity (Diehl et al., 2012; Granier et al., 2011; Lamarque et al., 2010; van der
Werf et al., 2006). The overall emission trends (black lines) illustrate a decrease of anthropogenic emissions
since 1900 (Figure 8a) and a high variability of biomass burning emissions over the whole period (Figure 8b).
For anthropogenic emissions, the largest BC emissions in EEU and CEU regions occurred in 1980, followed by
decreasing trends. WEU had the strongest BC emissions lasting until 1960, followed by a decrease of BC
emissions lasting the present-day. In 2008, anthropogenic BC emissions in region EEU, CEU and WEU are
comparable with an order of 0.2 Tg yr$^{-1}$.
To investigate factors controlling long-term rBC trends preserved in the ELB ice core, the temporal evolution of
measured ice core rBC particles can be directly compared with that of atmospheric BC load at the ELB site, at
least in relative manner. This comparison is provided in Fig. 9, in which ice core record is averaged along a
decadal scale to be comparable with the historic BC emission data available on decadal scale only (Lamarque et
al., 2010). Specifically, we coupled the BC emission intensities in each region and their relative contribution to
the entire footprint area of ELB site (Figure 8c and d). The decadal BC emission burden in each region (Figure
8a and b) is therefore multiplied by the contribution of footprint density (Figure 7). Assumption behind this





comparison is that (i) the atmospheric circulation and transport patterns do not change with time and (ii) that the
mechanisms for BC depositing to snow remained constant. Hence, the proportionality between BC mass
concentration in snow and atmospheric BC load has not varied with time.
For summertime (JJA case, Fig. 9a)  optimal agreement in trend pattern is observed between the ice core rBC
and the atmospheric BC estimated in the lower 2 km layer with an increase at the onset of the 20$^{th}$ century and a
subsequent decrease since ~1980 ("best scenario"). Specifically, substantial increase in atmospheric BC load is
observed for the period 1910-1970, similar to the ELB rBC ice core record, only when the atmospheric BC
considers BC particles transported in the lowest 2 km layer of the atmosphere. On the other side, the estimation
derived from the entire atmospheric column does exhibit a different pattern. This comparison indicates that
changes primarily in European anthropogenic BC emissions (e.g., industry, traffic and residential combustion),
particularly ones of Eastern part of Europe, are consequently reflected in the ELB ice core rBC variability over
the last century.
For wintertime (DJF case, Fig. 9b), the ice core rBC variability before 1980 can be explained by the atmospheric
BC load (anthropogenic only) in the entire atmospheric column but without North American (NAM)
contribution. With NAM contribution included in the simulation, the atmospheric BC is overestimated before
1980 resulting in a flat or a slightly downward trend for the period 1910-1970, unlike to the ice core rBC trend.
However, the good agreement between long-term rBC changes of Greenland ice core and modeled BC
deposition in Greenland using a chemistry-climate model with an input of ACCMIP BC inventory confirm that
BC emission estimates for NAM from the ACCMIP inventory correctly quantify anthropogenic BC emissions in
North America (Lamarque et al., 2010). Consequently, the observed overestimation of NAM contribution for
winter at the ELB site (Fig 9b) is likely due to an overestimation of NAM footprint density in the statistical
process applied on FLEXPART simulation data. Finally, the estimated BC without NAM contribution is defined
as the "best scenario" for winter time.
Despite the similar features between the ice core rBC record and the best scenario for the atmospheric load
which support that anthropogenic BC increase in the 20$^{th}$ century is reflected in the ELB record, BC maximum
time period is not in total agreement (Fig. 9a and b). Unlike the ice core rBC that already largely increased in
1960 and peaked in 1970 for both summer and winter, the atmospheric BC load remarkably increases only in
1980. Substantial BC increase of ELB and Western European (CDD and CG) ice cores since the mid-20th
century reveals that BC emissions increased during that period at a wide regional European scale. In addition, the
CDD record shows a large increase in sulfate concentration since the mid-20th century lasting until ~1980
(Preunkert and Legrand, 2013; Preunkert et al., 2001). Knowing that sulfate and BC are often co-emitted in
anthropogenic emission sources, e.g., in industrial sectors, one can expect a large increase in European BC
emissions in 1960-1980, as suggested by the ELB ice core rBC record. The reliability of historic emission
inventories for BC is reported to be lower than for $SO_2$, CO and NOx emissions, particularly for the period prior
to 2000 (Granier et al., 2011), which is due to the uncertainties on BC emission factors for coal, gasoline and
diesel fuels in various sectors (differ by a factor of 10 or more in literatures) and activity data (Granier et al.,
2011; Vignati et al., 2010). Thus, the lack of substantial increase in the atmospheric BC load for the period
1960s-1970s could be associated primarily with underestimated European anthropogenic BC emissions for this
period (Fig. 8c and d).
Moreover, the ice core rBC record and the atmospheric BC load do not exhibit similar patterns after 1980.



Decreasing rates of the ice core rBC are much slower after 1980 onward for both seasons than the atmospheric
BC load (Fig. 9a and b). Furthermore, the summer rBC trend of the ELB ice core even increased since 2000,
although such a trend cannot be reported conclusively for winter layers (Fig. 4). The recent economic growth in
Eastern, and some part of Central, European countries (World Bank Group, 2016) can contribute to the
enhancement in the release of BC and co-emitted pollutants. Some of Eastern European countries have kept
increasing their sulfur emissions mainly from heat production and public electricity from 2000 onward (Vestreng
et al., 2007). Thus, the increase in rBC deposition at the Elbrus site, mostly identified in summer, was probably
related to enhanced emissions from anthropogenic sources located in Eastern and Central Europe. On the other
side, many of Eastern European countries, such as Ukraine and European part Russia which are geographically
close to the Mt. Elbrus, are the countries with the greatest land use for agriculture in Europe (Rabbinge and van
Diepen, 2000), and thus emissions of smoke aerosols from their agricultural waste burning are expected to be
significant in summer time (Barnaba et al., 2011; Bovchaliuk et al., 2013; Stohl et al., 2007). Large emissions of
smoke aerosols over Eastern Europe from summer forest/agricultural fires have been recently reported (Barnaba
et al., 2011; Bovchaliuk et al., 2013; Sciare et al., 2008; Yoon et al., 2011; Zhou et al., 2012) and burned area
from Global Fire Emissions Database (GFED) (Giglio et al., 2010) increased over Eastern Europe for the period
2004-2008 (Yoon et al., 2014). These emissions of smoke aerosols in the Eastern part of Europe may have
contributed to the observed summer BC increase in the ELB ice cores. Thus, the recent trend of the ELB ice core
rBC turning upward probably indicates changes in both anthropogenic emissions and summer forest/peat fires
over Eastern part of Europe in 2000s, which is not well reflected in the inventories.
Given the large existing uncertainties in historic BC emission inventories available to date, our rBC record
reconstructed from a high-altitude Caucasus ice cores should be useful to better constrain BC emissions.
Specifically, our study highlights the need for improving BC emission inventories from the Eastern part of
Europe since 1960. Reliability of Western European BC emissions could be more specifically assessed by
investigating high-resolution BC records extracted from Western European ice cores that would be more
representative of Western European emissions.

## 4 Conclusions

A high-resolution rBC record reconstructed from ice cores drilled from a high-altitude Eastern European site in
Mt. Elbrus (ELB), Caucasus, reported for the first time the long-term evolutions of rBC mass concentrations and
size distributions in the European outflows over the past 189 years, i.e., between year 1825 and year 2013. The
rBC record at ELB is largely impacted by rBC emissions located in the Eastern part of Europe. A large temporal
variability in rBC mass concentration was observed at both seasonal and annual timescales. This record is also
unique to document long-term variability of BC in this region of Europe.
In the first-half of 20th century, rBC concentrations increased by a 1.5-fold than its level in the preindustrial era
(before 1850). The rBC concentrations increased by a 5-fold in 1960-1980, followed by a decrease until ~2000
and a slight increase again since ~2000. Consistent increase in background levels, since the beginning of 20th
century, highlights that rBC background level in the atmosphere at ELB was also significantly altered. We have
also investigated the potential of size distributions of rBC particles in the ice cores as new a proxy to bring
additional information on rBC removal processes, seasonal transport patterns, and emission sources.



We simulated the atmospheric load of BC aerosols which were transported from the European continent, mainly
Eastern part of Europe, by coupling transport simulations (FLEXPART) to 20th-century BC emission inventories
(ACCMIP and MACCity). Similar features were observed between the ELB ice core rBC mass concentration
record and the best scenario for the atmospheric BC load at the ELB site: a BC increase at the onset of the 20th
century and a subsequent decrease since ~1980. This estimation evidently supports that European anthropogenic
activities resulted in the BC increase over Europe since ~1900, which was also seen in elemental carbon (EC)
records of Western European ice cores (Legrand et al., 2007; Thevenon et al., 2009). However, some
disagreements were seen between the ELB ice core rBC and the best scenario for atmospheric BC load at ELB,
e.g., (i) the lack of strong increase in the best scenario for the period 1960s and 1970s, unlike the ice core record,
(ii) the different decreasing rates after 1980 and (iii) the slightly increasing trend of the rBC ELB ice core record
that was not shown in the estimation. An explanation for such discrepancy could be that rapid enhancement of
BC emissions over Europe since 1960 and the recent BC changes in the Eastern part of Europe may not be well
accounted for in the emission inventories.
Most atmospheric BC measurements have focused on western and northern Europe (e.g., McMeeking et al.,
2010; Reche et al., 2011; Reddington et al., 2013) despite of growing evidences of strong aerosol emissions in
the Eastern part of Europe (Asmi et al., 2011; Barnaba et al., 2011; Bovchaliuk et al., 2013). It is thus critically
important to deploy new studies (atmospheric monitoring and investigation of ice archives) with a more
comprehensive European view, including both Western and Eastern areas. We suggest that century-long ice cores
at multiple high-altitude European sites with a homogeneous or well crossed- compared measurement techniques
are needed to better constrain past BC emissions, infer efficiency of present BC emission regulation, and help
establishing future regulations on BC emissions.

**Acknowledgments**
This work was supported by the PEGASOS project funded by the European Commission under the Framework
Programme 7 (FP7-ENV-2010-265148) and by the Russian Foundation for Basic Research (RFBR) grants 07-
05-00410 and 09-05-10043. This work received funding from the French ANR programs RPD COCLICO
(ANR-10-RPDOC-002-01) and the European Research Council under the European Community's Seventh
Framework Program FP7/2007–2013 Grant Agreement n° 291062 (project ICE&LASERS).  S. Lim
acknowledges support of the Korean Ministry of Education and Science Technology through a government
scholarship and of the Basic Science Research Program through the National Research Foundation of Korea
(NRF) funded by the Ministry of Education (2015R1A6A3A01061393). V. Mikhalenko and S. Kutuzov
acknowledge support of the Russian Academy of Sciences (Department of Earth Sciences ONZ-12 Project) and
RFBR grant 14-05-00137. Grateful thank to M. Zanatta for technical help in SP2 operation, S. Preunkert for
technical help in ice core cutting, M. Legrand for helpful discussions, A. Berchet and J-L Bonne for help in
FLEXPART simulations and N. Kehrwald for analytical help.

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





**Table and figures**

**Tables**
**Table 1. rBC mass concentrations at seasonal resolution and relative increases compared to 1825-1850**
**(preindustrial era) for different time periods.**

| Time period | Summer | | Winter | |
|---|---|---|---|---|
| | Concentration in µg L$^{-1}$ (median ± SD) | Relative increase to 1825-1850 | Concentration in µg L$^{-1}$ (median ± SD) | Relative increase to 1825-1850 |
| 1825-1850 | 4.3±1.5 | 1.0 | 2.0±0.9 | 1.0 |
| 1850-1900 | 5.3±2.6 | 1.1 | 2.5±1.4 | 1.0 |
| 1900-1950 | 7.9±3.9 | 1.5 | 3.2±1.6 | 1.4 |
| 1950-2000 | 20.0±7.1 | 4.3 | 6.0±2.7 | 2.7 |
| 1960-1980 | 22.6±7.2 | 5.0 | 7.1±2.5 | 3.3 |
| 2000-2013 | 17.7±5.9 | 3.9 | 5.4±2.3 | 2.4 |





























**Figure captions**

**Figure 1.** Location of the ice core drilling site (43°20'53, 9"N, 42°25'36, 0"E, 5115 m a.s.l., indicated by the red star or arrow) in the Mt. Elbrus, the western Caucasus mountain range between the Black and the Caspian seas.

**Figure 2.** Five regions classified as potential rBC emission sources regions.

**Figure 3.** A Profile of high-resolution rBC concentration of Mt. Elbrus ice cores. (a) whole rBC profile of both the 2013 core and the 2009 core, and (b) the 2009 core from top to 20 m corresponding to the blue region in (a). In (b), lower resolution (at ~5-10 cm resolution; black color) and high resolution (at 1 cm resolution; red color) rBC profiles obtained from discrete analysis and continuous flow analysis, respectively, are shown. For a whole rBC record, a section of lower-resolution signals of the 2009 core (corresponding to calendar year 2009) was replaced with the high-resolution rBC signals of the 2013 core. Gray text on top of figures stands for calendar year corresponding to ice core depth.

**Figure 4.** Annually averaged temporal evolution in rBC mass concentration of the ELB ice cores. (a) Summer and (b) winter. Thin solid line is medians and dashed lines are lower and upper 10[th] percentiles of the seasonal rBC values. Upper 10[th] percentiles do not exceed 75 $\mu$g L[-1] and 35 $\mu$g L[-1] for mid-summer and mid-winter, respectively. Thick lines are 10-year smoothing of medians. Discontinuous thin lines indicate ice layers with unclear seasonality or unanalyzed ice layers. Note different y-scales for seasonal rBC concentrations.

**Figure 5.** Time series of mass mode diameter (MMD) of seasonal rBC size distributions for the period of 1940-2009. The MMD was obtained by fitting a log-normal curve to the measured distribution. Horizontal lines stand for geometric means for summer (red) and winter (blue).

**Figure 6.** Air mass footprint area for (a) June to August (JJA) and (b) December to February (DJF) in the atmospheric column and (c) JJA in the lowest 2 km in the atmosphere. Color bar on the left indicates footprints density with a process defined unit (p.d.u.). The location of the ELB site is marked by a white triangle. JJA and DJF correspond to summer and winter of the ELB ice core depth, respectively.

**Figure 7.** Contribution of each regional footprint density (%) for (a) JJA and (b) DJF in the atmospheric column and (c) JJA in the lowest 2 km in the atmosphere. Footprint density of each region is divided by the footprint density of the entire footprint area (EEU+CEU+WEU+NAF+NAM+Others) and then described in percentage. Information for each region is found in the Sect. 2.4.

**Figure 8.** Historic regional BC emissions and atmospheric BC load at ELB for the period 1900-2008. In (a) and (b), anthropogenic and biomass burning (forest fires and savanna burning) BC emissions estimated by ACCMIP and MACCity (Diehl et al., 2012; Granier et al., 2011; Lamarque et al., 2010; van der Werf et al., 2006). In (c) and (d), atmospheric BC load (Tg yr[-1]) is calculated by multiplying decadal-scale BC emissions in each region (a and b) by its relative contribution to the entire footprint area of ELB site (figure 7). In (c), both anthropogenic and biomass burning emissions are used for the reconstruction in JJA, as this type of biomass burning (forest fires and savanna burning) is the most frequent in summer and in (d), only anthropogenic emissions are used for DJF. Details are found in the text.

**Figure 9.** Comparison in temporal evolution between the rBC mass concentration of the ELB ice core and the estimates of atmospheric BC load at the ELB site, on a decadal scale. (a) JJA and (b) DJF. Best scenarios for atmospheric BC load are shown in black thick lines. In (b), NAM stands for North America. See the text and Figure 8c and d for calculations of the atmospheric BC load.





**Figures**

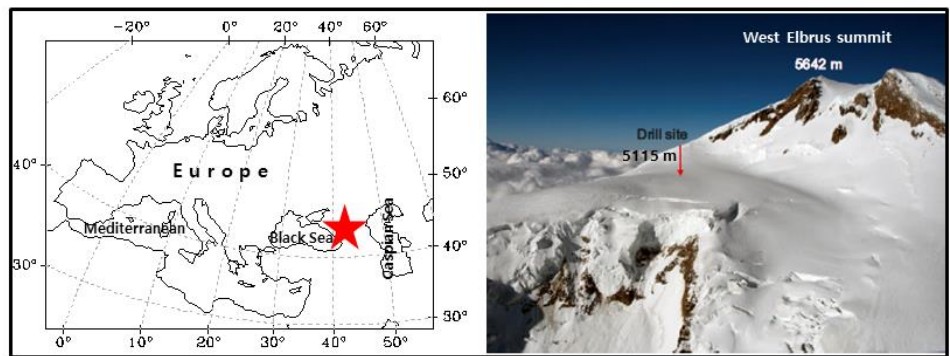


**Figure 1**





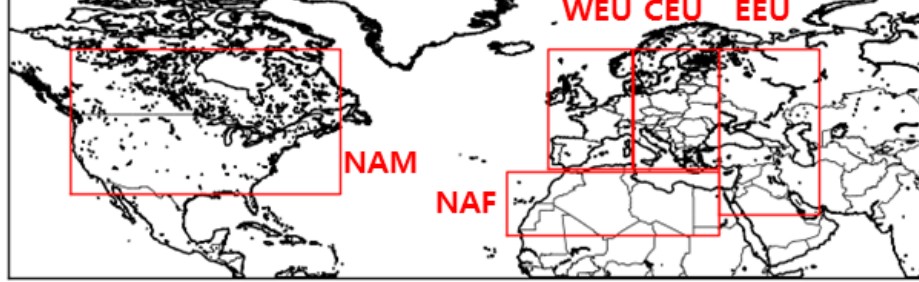


**Figure 2**






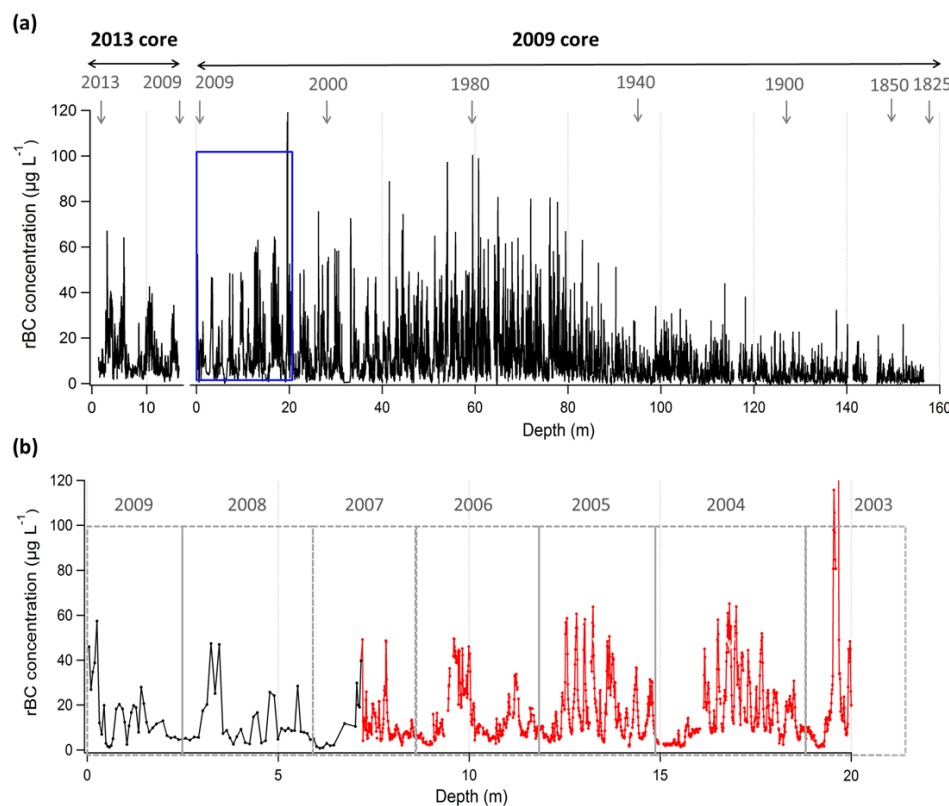


**Figure 3**



















**(a)**

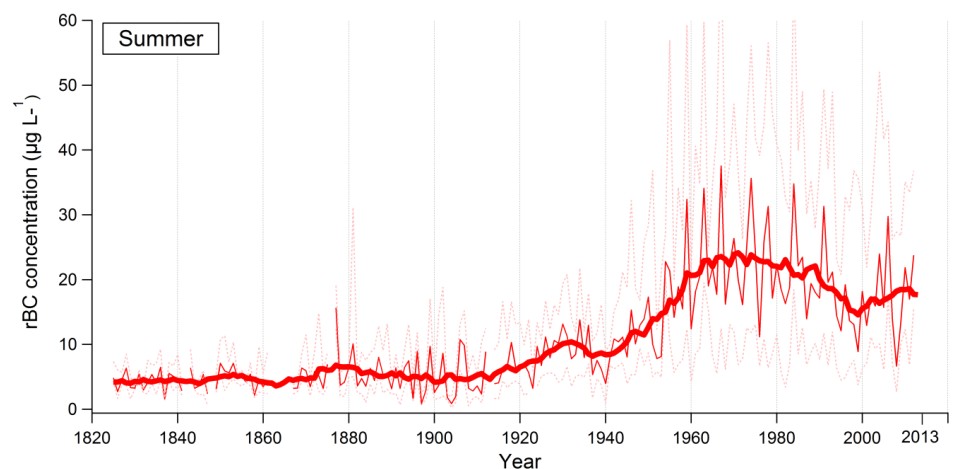

**(b)**

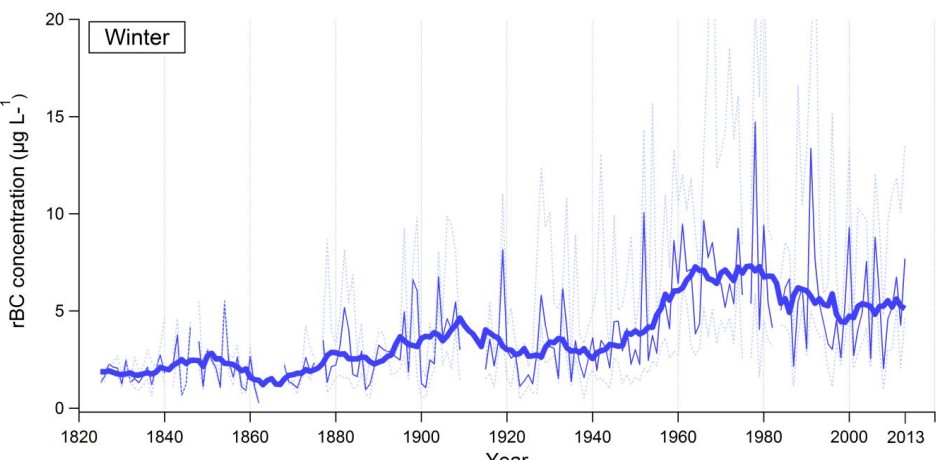

**Figure 4**



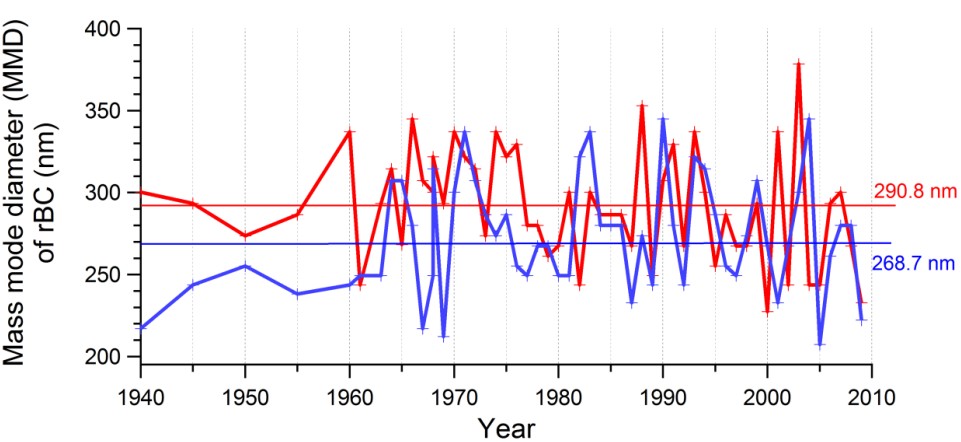


**Figure 5**






























**(a)**                                    **(b)**
**(c)**
**Figure 6**





**(a)**

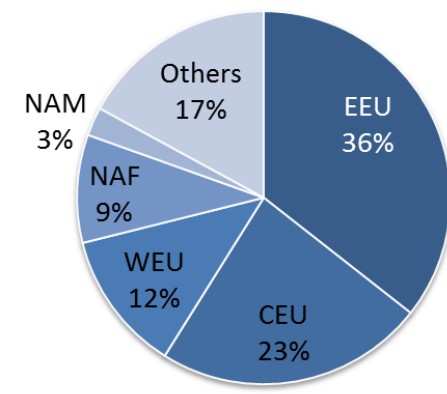

**(b)**

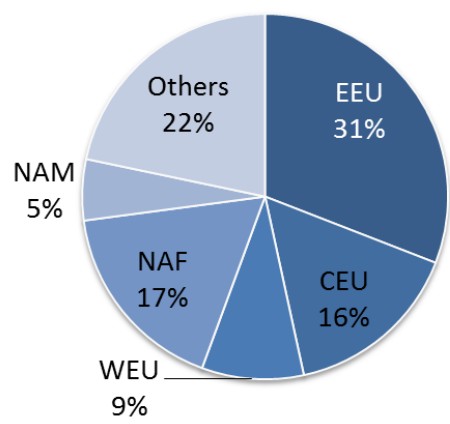




**(c)**

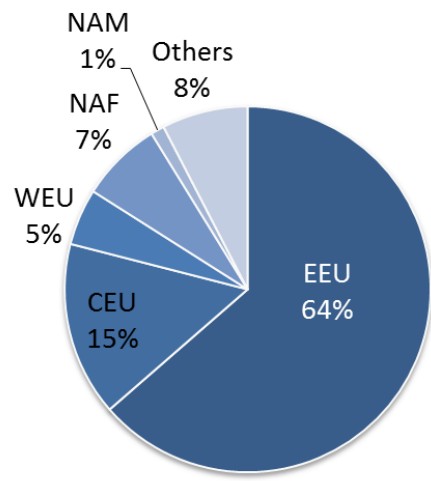


**Figure 7**




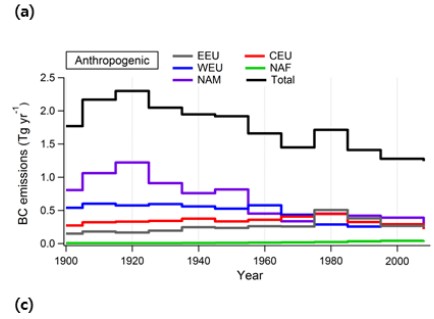

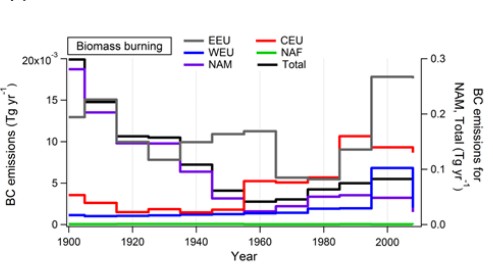

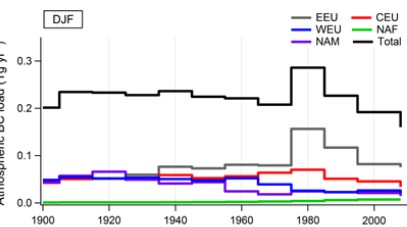


**Figure 8**




**(a)**

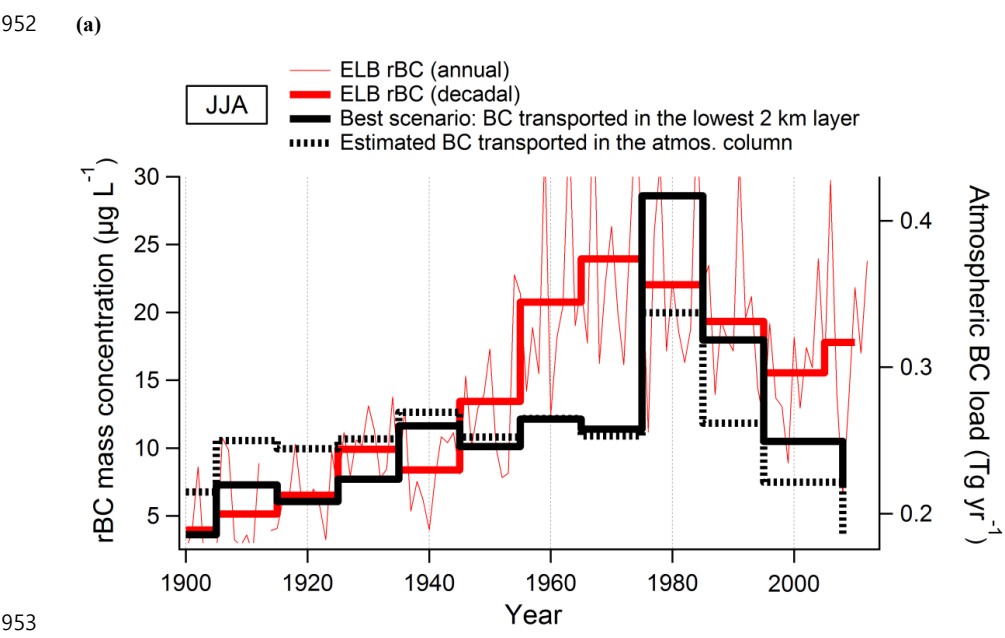

**(b)**

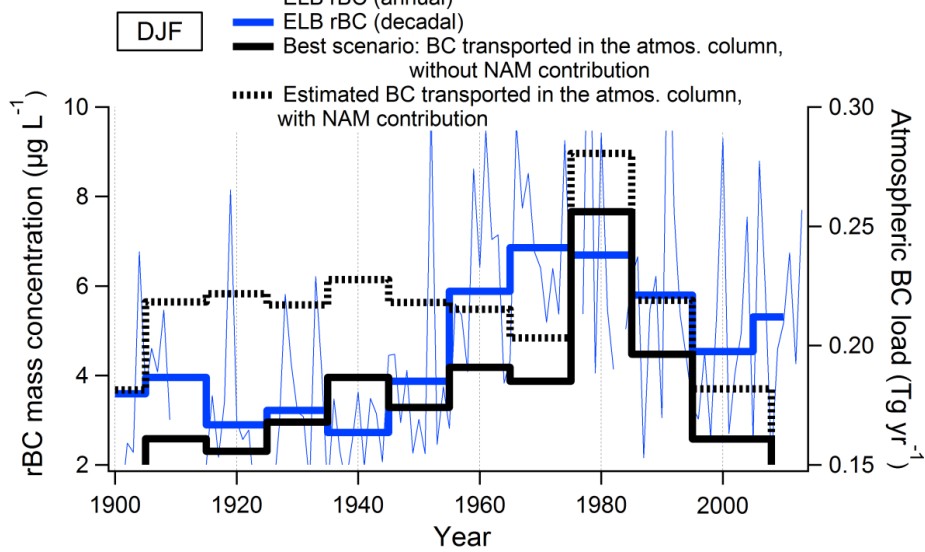

**Figure 9**



