# Peer review of "Black carbon variability since preindustrial times in Eastern part of Europe reconstructed from Mt Elbrus, Caucasus ice cores"

_Atmospheric Chemistry and Physics, 2016_

## Referee Comment (RC1) · Anonymous Referee #1 · 14 Oct 2016

This paper presents the first high-resolution record of refractory BC (rBC) from Eastern Europe based on an ice core retrieved from Mt. Elbrus and covering the time period 1825-2013. The trend in rBC concentration is discussed with respect to atmospheric BC loads simulated with the FLEXPART model using BC emission inventories. The main conclusion is that the record mainly reflects BC emissions in the Eastern part of Europe. Discrepancies between the ice core record and the simulated BC load suggest an underestimation of anthropogenic and biomass burning BC emissions in Eastern Europe. The Mt. Elbrus rBC record has the highest temporal resolution of the BC or EC records, which have been published from European ice cores (Col du Dome, Colle Gnifetti, Fiescherhorn, all in the Western part of Europe). The authors made a

real effort to attribute potential source areas, instead of just presenting the record. This is very valuable and the study nicely demonstrates the importance of such records to constrain highly uncertain BC emission estimates. The paper is generally clear, well written, thorough, and suitable for ACP, and should be published.

However, I have some comments that the authors might want to consider for the sake of clarity in the paper.

I know it is kind of tradition of the "Grenoble" group to classify the ice core data into summer and winter values and there might be circumstances were this is justified. However, this is always difficult, since the record does not contain clear time markers for the seasons and therefore assumptions have to be made. In this case, the 25th and 75th percentiles of thickness of an annual layer were arbitrarily chosen, assuming equal distribution of precipitation (and preservation on the glacier) throughout the year. This is conducted without explaining the hypothesis behind. The obtained summer and winter rBC records do show similar trends and the slight differences around 1900 are not discussed at all. As expected the JJA and DJF scenarios for the atmospheric BC load are different, with much higher contributions from North America (NAM) to DJF. However, the authors do not question their summer and winter classification in the ice core because of this finding, but do instead explain the difference with an overestimation of the NAM footprint density by the simulation. To my opinion, the classification into JJA and DJF needs better justification, for example by showing that the annual ice core rBC concentrations and annual atmospheric BC loads agree less. What is puzzling is that in the other manuscript about this ice core (Kozachek et al., CPD 2016), classification into seasons is conducted by introducing the mean delta18O value as threshold. If at all, the procedure should be the same. Please also reconcile the details about the core (20.4 m here, 20.5 m in Kozachek et al.; dating uncertainty few years here, +- 1 year in Kozachek et al.) and include a reference to that manuscript.

The rBC size distribution data are very valuable since they support other findings that the rBC particle sizes in snow and ice are larger than in the atmosphere. However, the

difference in MMD between summer and winter (Fig. 5) is not so obvious to me. The main discrepancy is for the few data points before 1960 where the data coverage is anyway poor. Have you tested if the MMDs for the period after 1960 are significantly different, considering the strong variability of the data?

Minor comments:

The English is generally good, but still some editing is required.

Line 63: rephrase: that is reconstructed in the downstream of Europe.

Line 95: Please specify "upper section" (move this up from line 116).

Line 117: Give max and min numbers of data points per year.

Lines 211-214: The fact that biomass burning emissions frequently occur in summer should be reflected in the emission estimates. I do not understand the argument for not considering the biomass emissions in DJF.

Lines 232-236: Include Jungfraujoch (e.g. Bukowiecki et al., Aerosol and Air Quality Research, 16: 764–788, 2016).

Line 293: Mikhalenko et al. (2015) do not mention aerosol removal processes. Please clarify that you assume that wet deposition dominates, since there is often and regular precipitation throughout the year.

Line 355: Matthias, 2004: Is this Matthias and Bösenberg, 2002?

Figure 2 would benefit from a better quality map. Please indicate location of ELB and explain abbreviations in the figure (NAM etc).

Fig. S1: The overlap between the 2009 and 2013 cores is not convincing. Could you support this with other ice core parameters (e.g. stable isotopes)?

---

## Referee Comment (RC2) · Anonymous Referee #2 · 18 Oct 2016

This paper shows the rBC concentration profile retrieved from a long ice core and a shallow core drilled on Mt. Elbrus, Caucasus, covering the period 1825-2013. This rBC profile has, by far, the highest temporal resolution compared to the OC and EC profiles obtained from ice cores from the Western European Alps (Col du Dome, Colle Gnifetti and Fiescherhorn). The author compares the rBC profiles with the atmospheric BC load at the drilling site computed with the FLEXPART lagrangian particle dispersion model: the available European BC emission inventories and the BC potential sources areas are used to compute the estimated BC load. BC emissions from the Eastern Europe appeared to be the main source of BC for Mt. Elbrus. The differences between the measured BC profile and the estimated atmospheric load are suggested to be caused

by an underestimation of the real emissions in the inventories; therefore, the author suggests that the retrieved rBC profile may be used as a new valuable constrain for the past Eastern European emission inventories. Mid-latitude ice cores are extraordinary environmental and climatic archives and this work represents a great step forward in the past atmospheric aerosols' reconstructions. The structure of the paper is clear and the modeling efforts contribute to the interpretation of the ice core based rBC profile. This paper is really well done and the argument is in accordance with the aim of the ACP Journal, therefore I suggest publishing it.

I have two comments, which the author might want to consider:

1) I agree with the comment of Referee #1 about the summer/winter layers subdivisions. If possible, I will recommend reinforcing the rBC-based annual layers determination with some other seasonally varying parameters, such as water stable isotopes, thus being in agreement with the other paper about the Elbrus ice core (Kozachek et al., CPD 2016).

2) I have found the rBC particles' MMD time series and the related interpretation very interesting and promising. I agree with all the interpretations but, however, the seasonality is not clear since the 1960s; particularly, during the 1980s the winter MMDs are even larger than the summer ones. I don't think that the difference between summer and winter is statistically significant in the period 1960-2010, can you please add some comments and interpretations about that? Or at least describe the MMD time series more in details.

Minor comments:

Line 37: it's better to write: "to be transported" instead of "to transport".

Line 38: "In high-altitude or –latitude areas ": missing word?

Line 39: "that may accelerate": it's better to write, "in accelerating".

Line 47: "proximity": how much? It's better to specify for the sake of clarity.

Line 50: please add a phrase regarding the BC/EC relation and write that there aren't other rBC records in this region.

Line 102: "Nd YAG laser": please write "Nd:YAG laser", with colon.

Line 123: "single rBC", I will add "particle".

Line 198: you may want to underline that the procedure is the same as for the entire atmospheric column.

Line 228: try to be clearer, e.g. "The highest rBC mass concentrations were observed..."

Line 236: substitute "consistent to..." with "consistent with..."

Line 259: if you write and compare the absolute values for the EC with you rBC analyses it will be better to write something about the conversion factor also in this part of the paper, or at least specify "how" to compare the values explicitly.

Line 293: please clarify why dry deposition is not playing a significant role in the rBC particles diameter changing.

Line 295: should surface snow melting modify the rBC size distribution? Explain and add references.

Line 332: can you exclude the surface snow melting effect in increasing the rBC MMD in the 2003 summer layer? Please explain.

Line 386: "BC depositing to snow": "BC depositing ON snow".

Line 462: "as new a proxy": write "as a new proxy".

For what concern the figures I would personally prefer having the deepest and the oldest parts always on the right or on the left (but this is up to you).

---

## Author Comment (AC1) · 30 Nov 2016

We would like to thank the two anonymous referees for their careful reading of the manuscript, and also the time they dedicated to evaluating this study. All comments were highly insightful. Please find below our point-by-point response to the critiques and a highlight to the changes made to the manuscript to address these. We strongly feel that we were able to address all the points raised.

1. I know it is kind of tradition of the "Grenoble" group to classify the ice core data into summer and winter values and there might be circumstances were this is justified. However, this is always difficult, since the record does not contain clear time markers for the seasons and therefore assumptions have to be made. In this case, the 25th

and 75th percentiles of thickness of an annual layer were arbitrarily chosen, assuming equal distribution of precipitation (and preservation on the glacier) throughout the year. This is conducted without explaining the hypothesis behind.

: We agree with the reviewer's point that classifying the ice core data into summer (or warm season) and winter (or cold season) values was done without clear time markers. However, the ELB ice cores show very clear seasonality with the high summer values and the low winter values of water stable isotopic composition (d18O and dD) and NH4+. Both d18O and NH4+ were thus used to classify seasonal ice layers of the cores (summer-half year and winter-half year). We think our proposed method for dating and seasonality is the best one for the moment. We also agree with the reviewer's point that the 25th and 75th percentiles of thickness of a seasonal snow layer were chosen in this study assuming equal distribution of precipitation through a year and the hypothesis behind choosing this method is not clearly explained. In the Caucasus region, most of the annual precipitation occurs in the western and southern sections of the Caucasus, reaching 3240 mm y−1 at Achishkho weather station (1880 m). Precipitation ranges between 2000 and 2500 mm y−1 at 2500 m a.s.l. in the west and declines to 800–1150 mm y−1 in the east on the northern slope of the Caucasus (Mikhalenko et al., 2015). A regular year round precipitation has been observed in the Western Caucasus at Klukhorskiy Pereval station (2037 m a.s.l., 50 km westward; the location is indicated with number "7" in Kozachek et al., 2016, Fig. 1), where the proportions of mean summer and winter precipitations are 0.94 m (52%) and 0.87 m (48%), respectively, and precipitation of each month accounts for 6-11 % of total precipitation for the period 1966-2009 (www.meteo.ru). In our 2009 ELB core, the mean annual snow accumulation rate (1455 mm w.e. y-1 for the last 140 years) obtained by counting annual layers suggests that the deposited snow at the ELB site is well preserved without significant loss driven by wind erosion, although direct precipitation measurements are not available at the drilling site. Seasonal snow is also well preserved, with nearly equal deposition amounts from the warm season (45% of total accumulation) and the cold season (55% of total accumulation), e.g., a short firn core

spanning the years 2012-2009 (Kutuzov et al., 2013). Hence, we suggest that it was reasonable to assume that precipitation at the ELB site is equally distributed through a year. Furthermore, our separation method of seasonal snow layers is supported by the coincidence of maximum (or minimum) values of both d18O and NH4+ in the annual snow layers. Most of maximum (or minimum) values of both d18O and NH4+ were observed in the 25th and 75th percentiles of thickness of a summer (winter) snow layer. However, we also observed unusual shift in NH4+ from d18O pattern, although it was rarely observed (roughly 10-year-ice layers over the entire 198-year-long record). The impact of inaccurate seasonal separation on rBC was limited by calculating median rBC mass seasonal concentration values. We finally concluded that the 25th and 75th percentiles of thickness of a seasonal snow layer were chosen in this study assuming equal distribution of precipitation through a year and that summer or winter rBC mass concentrations that were provided following this method are corresponding to summer maximum or winter minimum values of climate proxies in an annual layer.

We added a following sentence in line 150: "This seasonal separation method is fairly supported by the fact that (i) observed precipitation in the Western Caucasus is equally distributed throughout a year (e.g., at Pereval Klukhorskiy observatory which is located at 2037 m a.s.l. and only 50 km from the drilling site: 52% of the annual precipitation (resp. 48%) is observed during summer (resp. winter) and each monthly precipitation accounts for 6-11 % of total precipitation for the period 1966-2009; www.meteo.ru) and (ii) maximum or minimum values of both d18O and NH4+ coincide for most of the Elbrus core annual ice layers."

2. The obtained summer and winter rBC records do show similar trends and the slight differences around 1900 are not discussed at all.

: As indicated by the reviewer, we observed slight increase of winter rBC values in 1900-1920 with respect to summer rBC values. We do not fully understand why the seasonal differences were shown in the period 1900-1920. There might be increased winter BC inputs for the period transporting to the ELB site. We used BC emission

inventory data of Lamarque et al. (2010), which are well supported by rBC deposition reconstructed from Greenland ice cores (McConnell et al., 2007). In McConnell et al. (2007), rBC particles that transported from North America markedly increased at the beginning of 20th century, indicating increased BC emissions in North America for the period. The enhanced BC input from North America might be detected in the ELB ice core layers of the period 1900-1920, although it was not shown in our simulations.

The following sentence was added in line 254. "Meanwhile, the slight increase of winter rBC values in 1900-1920 with respect to summer rBC values are not well understood. Although speculative, it may reflect increased winter BC inputs transporting through the free troposphere (FT) from North America, where BC emissions markedly increased at the beginning of 20th century"

3. As expected the JJA and DJF scenarios for the atmospheric BC load are different, with much higher contributions from North America (NAM) to DJF. However, the authors do not question their summer and winter classification in the ice core because of this finding, but do instead explain the difference with an overestimation of the NAM footprint density by the simulation. To my opinion, the classification into JJA and DJF needs better justification, for example by showing that the annual ice core rBC concentrations and annual atmospheric BC loads agree less.

: Unlike scenario results for the seasonal atmospheric BC load at the ELB site, the seasonal rBC trends of the ELB ice cores were similar except for the periods 1900-1920 and 2000-2013. The winter rBC increased relatively for the first case and the summer rBC increased more obviously for the latter case. Both features were not shown in the simulations. Particularly, the increased winter rBC concentrations in ~1900-1920 may be linked with stronger BC inputs from North America at the beginning of the 20th century when the BC emissions were the strongest in that region. We do not understand well the mechanisms by which North American BC emissions could be strongly detected in the ELB ice core for this period. The following sentence was added in line 406. "Consequently, the observed overestimation of NAM contribution for winter

at the ELB site (Fig 9b) is likely due to an overestimation of NAM footprint density in the statistical process applied on FLEXPART simulation data, although the stronger BC inputs from NAM might have contributed to the increased winter rBC concentrations of the ELB ice core at the beginning of 20th century." Unfortunately, anthropogenic BC emissions from ACCMIP are available on the decadal scale only. We thus cannot show annual atmospheric BC load.

4. What is puzzling is that in the other manuscript about this ice core (Kozachek et al., CPD 2016), classification into seasons is conducted by introducing the mean delta18O value as threshold. If at all, the procedure should be the same. Please also reconcile the details about the core (20.4 m here, 20.5 m in Kozachek et al.; dating uncertainty few years here, +- 1 year in Kozachek et al.) and include a reference to that manuscript.

: Mikhalenko et al. (2015) has established age scale of the ELB ice core using NH4+ and succinic acid, and posteriori validation with d18O, resulting in a 2-year difference between annual layer counting of d18O signal and the NH4+ stratigraphy at 106.7 m. Kozachek et al. (2016) has made the ice core dating using annual d18O, d18O threshold, and use of NH4+ and succinic acid if issues with d18O. They initially reported a 1-year uncertainty of the dating but recently corrected this estimate to 2-years to agree with Mikhalenko et al. (2015)( see response to reviwers provided by Kozachek et al. (2016), TCD) . Finally, both Mikhalenko et al. (2015) and Kozachek et al. (2016) have reported that a 2-year difference between annual layer counting of d18O signal and the NH4+ stratigraphy at 106.7 m. This is an excellent agreement on age scales that were obtained by two methods, suggesting robust dating results of the ELB ice core from top to 106.7 m. In our study, we discussed rBC annual variability down to 156.6 m, corresponding to year 1825. the dating uncertainty from the surface to 106.7 m is 2-years as indicated by Mikhalenko et al. (2015) and Kozachek et al. (2016), but the uncertainty may be larger below 106.7 m due to ice thinning.

We corrected in line 141: "The dating uncertainty is 2-year between 106.7 m and the top (Kozachek et al., 2016; Mikhalenko et al., 2015) and probably larger below 106.7

m due to ice thinning" We corrected in line 72: ".. a 20.5 m-long ice core (the 2013 core)..."

5. The rBC size distribution data are very valuable since they support other findings that the rBC particle sizes in snow and ice are larger than in the atmosphere. However, the difference in MMD between summer and winter (Fig. 5) is not so obvious to me. The main discrepancy is for the few data points before 1960 where the data coverage is anyway poor. Have you tested if the MMDs for the period after 1960 are significantly different, considering the strong variability of the data?

: We agree with the reviewer's point that the difference in MMD between summer and winter is not very obviously shown. We thus conducted student's t-test on the MMD between summer and winter for the period 1960-2009 and the test resulted in significantly different mean MMD values between two seasons with $p < 0.01$. On the other hand, summer (or winter) MMDs for two periods for which highly variable rBC concentrations were observed, e.g., the period 1960-1999 and the period 2000-2009, were not significantly different. The following sentence was added in line 285: "No statistically significant temporal change in rBC MMD was identified over the 1940-2009 period." The following sentence was added in line 302: "The difference in seasonal rBC size distributions are statistically significant ($p < 0.01$)."

6. Line 63: rephrase: that is reconstructed in the downstream of Europe.

: The sentence revised as follow: "The ice core record therefore provides information on long-term variability and evolution of BC emissions of Europe."

7. Line 95: Please specify "upper section" (move this up from line 116).

: The "upper section" in the sentence was specified as follow: "The upper section of the 2009 firn core (surface to 7.2 m depth) was analyzed discretely."

The firn depth in line 221 was revised to 7.2 m.

8. Line 117: Give max and min numbers of data points per year.

: We added max and min numbers of data points per year in line 117: "The density of rBC data points per year (N=8~376) depends on annual snow accumulation rates and ice thinning with depth."

9. Lines 211-214: The fact that biomass burning emissions frequently occur in summer should be reflected in the emission estimates. I do not understand the argument for not considering the biomass emissions in DJF.

: We agree with the reviewer's point that the argument why biomass burning emissions were considered only for summer simulations is not clear. ACCMIP inventory (Lamarque et al., 2010) provides anthropogenic BC emissions on a decadal scale and biomass burning (savanna and forest burnings) BC emissions on a monthly scale. The figure below (Fig. 2 here), which is a part of Saehee LIM's PhD dissertation, shows that biomass burning BC emissions (kg/m2/s) in Europe that were calculated by ACCMIP were frequent in summer time and minimized in winter time. The biomass burning BC emissions in May to August are larger by two orders of magnitude than those in November to February.

This is now clearly stated in the sentence in line 213 as follow: " "We used anthropogenic emission only for constraining BC emissions in DJF and both anthropogenic and biomass burning emissions for constraining BC emissions in JJA, because seasonal biomass burning BC emissions are maximized in summer time (May to August), being two orders of magnitude larger than during winter time (September to February), as respect to anthropogenic emissions occurring year-round (Lamarque et al., 2010)."

10. Lines 232-236: Include Jungfraujoch (e.g. Bukowiecki et al., Aerosol and Air Quality Research, 16: 764–788, 2016).

: The reference (Bukowiecki et al., 2016) was added with relevant discussion in lines 232-236. In line 231, the following sentence is added. "In contrast to the boundary layer sites, the atmospheric measurements at high-elevation sites in Europe (e.g., Puy de Dôme at 1465 m a.s.l., Sonnblick at 3106 m a.s.l. and Jungfraujoch at 3580 m a.s.l.)

revealed 2 to 3 times higher EC levels during summer than winter (Bukowiecki et al., 2016; Pio et al., 2007; Venzac et al., 2009), "

11. Line 293: Mikhalenko et al. (2015) do not mention aerosol removal processes. Please clarify that you assume that wet deposition dominates, since there is often and regular precipitation throughout the year.

: We agree with the reviewer's point that Mikhalenko et al. (2015) did not mention aerosol removal processes and the line 293 in our manuscript can be misleading. We therefore corrected the sentence as follow: "The shift of rBC sizes induced by dry deposition should be negligible, as quite high (100-200 mm/month) and fairly constant precipitation rate throughout the year near the drilling site (e.g., 52% and 48% of annual precipitation observed in summer and winter, respectively, at Klukhorskiy Pereval station (2037 m a.s.l., 50 km westward; Kozachek et al., 2016) suggests that wet deposition can be the dominant aerosol removal pathway at this site."

12. Line 355: Matthias, 2004: Is this Matthias and Bösenberg, 2002?

: The reference "Matthias, 2004" in line 355 should be "(Matthias et al., 2004)" and the relevant reference info should be revised. Matthias et al. (2004) showed regular lidar observations of the vertical aerosol distribution at 10 European Aerosol Research Lidar Network (EARLINET) stations since 2000, for which they used the planetary boundary layer (PBL) height (km asl) at each station. Two mountain stations (L'Aquila at 1742 m agl and Potenza at 1536 m agl) showed monthly mean PBL height above 2 km asl and often higher weakly PBL height up to 3 km asl, while the PBL height was lower with monthly mean PBL of 1-2 km asl at the other stations. Thus, simulations for summer particle footprint within the lower 2 km layer in the atmosphere performed in our study are fairly consistent to the real PBL height at an area surrounding mountain and realistic aerosol transport to the drilling site.

13. Figure 2 would benefit from a better quality map. Please indicate location of ELB and explain abbreviations in the figure (NAM etc).

: Figure 2 (Fig 3. here) was replaced with a better quality map as follow. The location of ELB is indicated by a red circle symbol. The abbreviations were explained in the final version of the manuscript as below, but not in this comment due to limited space for figure caption.

Figure 2. Five sub-regions classified as potential rBC emission source regions. Elbrus drilling site (43°20'53,9"N, 42°25'36,0"E) is indicated by a red circle. WEU, CEU, EEU, NAF and NAM represent Western Europe, Central Europe, Eastern Europe, North Africa and North America, respectably.

14. Fig. S1: The overlap between the 2009 and 2013 cores is not convincing. Could you support this with other ice core parameters (e.g. stable isotopes)?

:An overlapping section (m w.e.) of the 2013 core and the 2009 core was described with water stable isotope (d18O) values in the following figure. The common d18O values were observed in the 2013-core depth of 6.8-10.7 m w.e., corresponding to year 2009-2007. The current Fig. S1 was replaced with the following figure (Fig 4 here).

Figure S1. An overlapping section of the 2009 core and the 2013 core. We used the common d18O feature dated year 2009-2007 and located at 7-11 m w.e. depth along the 2013-core depth scale to extend the 2009-core record (main core) with the 2013-core record.

[Figure]

[Figure]

**Fig. 1.** Map showing the region around Elbrus (black rectangle in the world's map in the lower right corner). Klukhorskiy Pereval station is indicated with number "7". (from Kozachek et al., 2016).

[Figure]

Figure I-10. Seasonality of BC emissions from biomass burning in Europe. The biomass burning BC emissions here include emissions from savanna burning and forest fires during the period 1900 to 2000. Mean monthly BC emissions with one standard deviation are estimated from ACCMIP BC emission inventory. Sources are from Lamarque et al. (2010).

**Fig. 2.** Seasonality of BC emissions from biomass burning in Europe during the period 1900 to 2000. Sources from Lamarque et al. (2010).

[Figure]

**Fig. 3.** Five sub-regions classified as potential rBC emission source regions. Elbrus drilling site (43°20'53,9"N, 42°25'36,0"E) is indicated by a red circle.

[Figure]

**Fig. 4.** An overlapping section of the 2009 core and the 2013 core.

---

## Author Comment (AC2) · 30 Nov 2016

We would like to thank the two anonymous referees for their careful reading of the manuscript, and also the time they dedicated to evaluating this study. All comments were highly insightful. Please find below our point-by-point response to the critiques and a highlight to the changes made to the manuscript to address these. For ease of discussion, we have continuously numbered the reviewer's comments. We strongly feel that we were able to address all the points raised.

15 I agree with the comment of Referee #1 about the summer/winter layers subdivisions. If possible, I will recommend reinforcing the rBC-based annual layers determination with some other seasonally varying parameters, such as water stable isotopes,

thus being in agreement with the other paper about the Elbrus ice core (Kozachek et al., CPD 2016).

: See reply to similar comment (#4) from reviewer #1. In addition, we added to the manuscript annual rBC variability (10th, 50th and 90th percentile values of annual snow layer; following figure) in Fig. 4c (Fig 1 here) with relevant description. The caption for Fig. 4c is as below.

Figure 4c. Annually averaged temporal evolution in rBC mass concentration of the ELB ice cores. Thin solid line is medians and dashed lines are lower and upper 10th percentiles of the annual rBC values. Thick line is 10-year smoothing of medians.

16. I have found the rBC particles' MMD time series and the related interpretation very interesting and promising. I agree with all the interpretations but, however, the seasonality is not clear since the 1960s; particularly, during the 1980s the winter MMDs are even larger than the summer ones. I don't think that the difference between summer and winter is statistically significant in the period 1960-2010, can you please add some comments and interpretations about that? Or at least describe the MMD time series more in details.

: See reply to similar comment (#5) from reviewer #1.

17. Line 37: it's better to write: "to be transported" instead of "to transport".

: It was rewritten to "to transport".

18. Line 38: "In high-altitude or –latitude areas ": missing word?

: Yes, "high" prior to "-latitude area" is missing. "In high-altitude or –latitude areas " in line 38 is thus revised to "In high-altitude or high–latitude areas ".

19. Line 39: "that may accelerate": it's better to write, "in accelerating".

: It was rewritten to "in accelerating".

20. Line 47: "proximity": how much? It's better to specify for the sake of clarity.

: We agree with the reviewer. European Alpine sites such as Col du Dôme, Colle Gnifetti and Fiescherhorn are approximately ~100 km away from big cities such as Lyon, France, Milano, Italy and Geneva, Switzerland, respectively. The sentence in line 47 was revised to "Particularly, the geographical proximity of the ice cores at high-altitude Alpine sites, e.g., European Alpine sites such as Col du Dôme, Colle Gnifetti and Fiescherhorn (Jenk et al., 2006; Legrand et al., 2007; Thevenon et al., 2009) to densely populated regions (approximately ~100 km) allows us to observe...".

21. Line 50: please add a phrase regarding the BC/EC relation and write that there aren't other rBC records in this region.

: This is an important point: we agree that terminology of BC derived from different methods should be differentiated. We thus add a sentence in line 53 as follow: "It should be noted that EC refers to data derived from thermal methods which are different than optical methods providing BC (including rBC derived from incandescence methods) (Petzold et al., 2013)". In line 63, we now stress that the ELB ice core rBC record is the first rBC record retrieved over Europe as follow: "For the first time, a high resolution, continuous rBC record is extracted from an ice core over Europe. The Elbrus rBC record thus brings new and unique information on long-term variability and evolution of BC European emissions."

22. Line 102: "Nd YAG laser": please write "Nd:YAG laser", with colon.

: It was revised.

23. Line 123: "single rBC", I will add "particle".

: "single rBC" was replaced with "single rBC particle" as recommended.

24. Line 198: you may want to underline that the procedure is the same as for the entire atmospheric column.

: This is a good point. We added a sentence in line 198: "..., while the simulation procedure is the same as for the entire atmospheric column".

25. Line 228: try to be clearer, e.g. "The highest rBC mass concentrations were observed: : :"

: For clarity, the sentence was revised to "The highest rBC mass concentration of an annual snow layer was observed in summer snow layer".

26. Line 236: substitute "consistent to: : :" with "consistent with: : :"

: It was revised as recommended.

27. Line 259: if you write and compare the absolute values for the EC with you rBC analyses it will be better to write something about the conversion factor also in this part of the paper, or at least specify "how" to compare the values explicitly.

: This is an excellent point: we agree that the most accurate and clear way to compare the absolute values for the EC of CDD and CG cores with for our rBC of ELB cores is describing corrected values based on existing lab experiments to evaluate different methods (Thermal (or thermal-optical) method vs. SP2-based incandescence method). Previously, Lim et al. (2014) conducted inter-comparison of the SP2-based incandescence method and thermal-optical method (EUSAAR2 protocol) for different field samples (i.e., Elbrus firn core, CDD snow fit, Greenland summit firn and Himalayan snow). In the experiments, Lim et al. (2014) found that thermal-optical method had disadvantages for providing accurate EC mass concentrations because (i) filtration efficiency, that is necessary prior to thermal-optical method, was strongly dependent on BC particle size and OC loading on the filter, (ii) presence of dust can cause negative EC artifact and (iii) OC pyrolyzation can bias OC/EC split point and then generally cause a positive EC artifact. On the other hand, the rBC results of SP2-based method was dependent on the SP2 gain setting that determines lower and upper rBC size limits. As a results, EC/rBC ratios were 0.5+/-0.2 for CDD snow pit and 1.0+/-0.4 for

ELB firn core, and the results came from mixed factors such as particle morphology and chemical composition of the field samples. We thus conclude that describing the comparison of our ELB rBC with previously measured EC of CDD and CG cores using corrected values or conversion factor may cause another uncertainty because we do not know BC or EC size and chemical composition (amount of OC and amount/type of dust) of the ELB, CDD and CG cores. Therefore we added a sentence in line 264 of the manuscript about why direct comparison of the ELB rBC with the EC of CDD and CG should be made with caution. We further added a reference, Lim et al. (2014) at the end of the sentence to guide readers.

28. Line 293: please clarify why dry deposition is not playing a significant role in the rBC particles diameter changing.

: Black Carbon particles are deposited in snow by either wet (i.e., in precipitation) or dry deposition. In general, BC removal from the atmosphere by wet deposition is estimated to be 3 times more efficient compared to dry deposition processes (Bauer et al., 2013). However, in some regions, dry deposition is considered to be the main process (or relatively more important than in other regions) for BC removal in the atmosphere (e.g., Khumbu valley in the Himalayas, Bonasoni et al. (2010); Yasunari et al. (2010). As discussed in replies to comment (#1) and (#11) we expect wet deposition to be the main deposition process at the ELB site, with an equal distribution along the year. Hence, it is reasonable to assume that BC deposition processes at the ELB site do not vary strongly along the seasons, and mainly involve wet removal by precipitation. We cannot quantify the proportion of dry deposited BC aerosols in snow, but this dry deposition effect should not be higher in specific month or season because observed monthly or seasonal precipitation rate is regular (e.g., at Klukhorskiy Pereval station). Once deposited, in addition to wind drift and erosion, particles can experience sublimation (snow to water vapor transition) or snow melt (Ginot et al., 2001; Schotterer et al., 2001). These postdeposition processes might affect BC concentrations and/or morphology within the snowpack. Only few studies have investigated how post-deposition

processes impact BC in snow (Hagler et al., 2007a, 2007b). Hagler et al. (2007b) showed relatively conservative behavior of EC in the 4-year snow pit layers at Summit, Greenland, where summer snow melting is limited similar to the ELB site, while water-soluble and –insoluble OC would undergo substantial post-depositional processing. Hence, it is reasonable to assume that post deposition processes are not impacting BC on snow at the ELB site.

29. Line 295: should surface snow melting modify the rBC size distribution? Explain and add references.

: To our knowledge, we do not know the studies of relationship between snow melting and rBC size modification. But there are plenty of studies showing that snow melt increase snow grain size. We first mentioned that ".. post-deposition processes are thus not expected to alter rBC size distributions." in line 295. We revised this sentence as follow: "Similarly, significant snow melt was not observed in the ELB summer ice layers. Although there is a lack of studies about the impact of snow melting on rBC size distribution, such processes would not be expected at the ELB drilling site"

30. Line 332: can you exclude the surface snow melting effect in increasing the rBC MMD in the 2003 summer layer? Please explain.

: The 2003 summer ice layer shows a clear shift on rBC MMD, which we attributed to influence of particle deposition from biomass burning plumes. This 2003 summer snow layer experienced some melting ( Kozachek et al., 2016), but we can rule out that such melting is driving the unusual MMD signal described above. We actually observed others snow layers with melting event (e.g., summer layers of year 2001 and year 2000), and all of these event did not show any anomalies of rBC MMD toward larger values.

31. Line 386: "BC depositing to snow": "BC depositing ON snow".

: It was corrected.

32. Line 462: "as new a proxy": write "as a new proxy".

: It was corrected.

33. For what concern the figures I would personally prefer having the deepest and the oldest parts always on the right or on the left (but this is up to you).

: We drew the figure by both methods, but finally decided the current one, because two of three papers that reported the alpine EC records (Jenk et al., 2006; Legrand et al., 2007; Thevenon et al., 2009) showed the figure of the past EC variability having the deepest parts on the left. We thus followed the method to help readers to compare their EC and our rBC records.

References

Bauer, S. E., Bausch, A., Nazarenko, L., Tsigaridis, K., Xu, B., Edwards, R., Bisiaux, M. and McConnell, J.: Historical and future black carbon deposition on the three ice caps: Ice core measurements and model simulations from 1850 to 2100, J. Geophys. Res. Atmos., 118(14), 7948–7961, doi:10.1002/jgrd.50612, 2013.

Bonasoni, P., Laj, P., Marinoni, A., Sprenger, M., Angelini, F., Arduini, J., Bonafè, U., Calzolari, F., Colombo, T., Decesari, S., Di Biagio, C., di Sarra, A. G., Evangelisti, F., Duchi, R., Facchini, M., Fuzzi, S., Gobbi, G. P., Maione, M., Panday, A., Roccato, F., Sellegri, K., Venzac, H., Verza, G., Villani, P., Vuillermoz, E. and Cristofanelli, P.: Atmospheric Brown Clouds in the Himalayas: first two years of continuous observations at the Nepal Climate Observatory-Pyramid (5079 m), Atmos. Chem. Phys., 10(15), 7515–7531, doi:10.5194/acp-10-7515-2010, 2010.

Bukowiecki, N., Weingartner, E., Gysel, M., Collaud Coen, M., Zieger, P., Herrmann, E., Steinbacher, M., Gäggeler, H. W. and and Baltensperger, U.: A Review of More Than 20 Years of Aerosol Observation at the High Altitude Research Station Jungfraujoch, Switzerland (3580masl), Aerosol Air Qual. Res., 16(3), 764–788, doi:10.4209/aaqr.2015.05.0305, 2016.

Ginot, P., Kull, C., Schwikowski, M., Schotterer, U. and Gäggeler, H. W.: Effects of postdepositional processes on snow composition of a subtropical glacier (Cerro Tapado, Chilean Andes), J. Geophys. Res. Atmos., 106(D23), 32375–32386, doi:10.1029/2000jd000071, 2001.

Hagler, G. S. W., Bergin, M. H., Smith, E. A. and Dibb, J. E.: A summer time series of particulate carbon in the air and snow at Summit, Greenland, J. Geophys. Res., 112(D21), D21309, doi:10.1029/2007JD008993, 2007a.

Hagler, G. S. W., Bergin, M. H., Smith, E. A., Dibb, J. E., Anderson, C. and Steig, E. J.: Particulate and water-soluble carbon measured in recent snow at Summit, Greenland, Geophys. Res. Lett., 34(16), L16505, doi:10.1029/2007GL030110, 2007b.

Jenk, T. M., Szidat, S., Schwikowski, M., Gaggeler, H. W., Brutsch, S., Wacker, L., Synal, H. A. and Saurer, M.: Radiocarbon analysis in an Alpine ice core: record of anthropogenic and biogenic contributions to carbonaceous aerosols in the past (1650-1940), Atmos. Chem. Phys., 6, 5381–5390, doi:10.5194/acp-6-5381-2006, 2006.

Kozachek, A., Mikhalenko, V., Masson-Delmotte, V., Ekaykin, A., Ginot, P., Kutuzov, S., Legrand, M., Lipenkov, V. and Preunkert, S.: Large-scale drivers of Caucasus climate variability in meteorological records and Mt Elbrus ice cores, Clim. Past Discuss., 1–30, doi:10.5194/cp-2016-62, 2016.

Kutuzov, S., Shahgedanova, M., Mikhalenko, V., Ginot, P., Lavrentiev, I. and Kemp, S.: High-resolution provenance of desert dust deposited on Mt. Elbrus, Caucasus in 2009–2012 using snow pit and firn core records, Cryosph., 7(5), 1481–1498, doi:10.5194/tc-7-1481-2013, 2013.

Lamarque, J.-F., Bond, T. C., Eyring, V., Granier, C., Heil, A., Klimont, Z., Lee, D., Liousse, C., Mieville, A., Owen, B., Schultz, M. G., Shindell, D., Smith, S. J., Stehfest, E., Van Aardenne, J., Cooper, O. R., Kainuma, M., Mahowald, N., McConnell, J. R., Naik, V., Riahi, K. and van Vuuren, D. P.: Historical (1850–2000) gridded anthropogenic

and biomass burning emissions of reactive gases and aerosols: methodology and application, Atmos. Chem. Phys., 10(15), 7017–7039, doi:10.5194/acp-10-7017-2010, 2010.

Legrand, M., Preunkert, S., Schock, M., Cerqueira, M., Kasper-Giebl, A., Afonso, J., Pio, C., Gelencsér, A. and Dombrowski-Etchevers, I.: Major 20th century changes of carbonaceous aerosol components (EC, WinOC, DOC, HULIS, carboxylic acids, and cellulose) derived from Alpine ice cores, J. Geophys. Res., 112(D23), D23S11, doi:10.1029/2006jd008080, 2007.

Lim, S., Faïn, X., Zanatta, M., Cozic, J., Jaffrezo, J.-L., Ginot, P. and Laj, P.: Refractory black carbon mass concentrations in snow and ice: method evaluation and inter-comparison with elemental carbon measurement, Atmos. Meas. Tech., 7(10), 3307–3324, doi:10.5194/amt-7-3307-2014, 2014.

Matthias, V., Balis, D., Bösenberg, J., Eixmann, R., Iarlori, M., Komguem, L., Mattis, I., Papayannis, A., Pappalardo, G., Perrone, M. R. and Wang, X.: Vertical aerosol distribution over Europe: Statistical analysis of Raman lidar data from 10 European Aerosol Research Lidar Network (EARLINET) stations, J. Geophys. Res., 109(D18), D18201, doi:10.1029/2004JD004638, 2004.

McConnell, J. R., Edwards, R., Kok, G. L., Flanner, M. G., Zender, C. S., Saltzman, E. S., Banta, J. R., Pasteris, D. R., Carter, M. M. and Kahl, J. D. W.: 20th-century industrial black carbon emissions altered arctic climate forcing, Science (80-. )., 317(5843), 1381–1384, doi:10.1126/science.1144856, 2007.

Mikhalenko, V., Sokratov, S., Kutuzov, S., Ginot, P., Legrand, M., Preunkert, S., Lavrentiev, I., Kozachek, A., Ekaykin, A., Faïn, X., Lim, S., Schotterer, U., Lipenkov, V. and Toropov, P.: Investigation of a deep ice core from the Elbrus western plateau, the Caucasus, Russia, Cryosph., 9(6), 2253–2270, doi:10.5194/tc-9-2253-2015, 2015.

Petzold, A., Ogren, J. A., Fiebig, M., Laj, P., Li, S.-M., Baltensperger, U., Holzer-Popp,

T., Kinne, S., Pappalardo, G., Sugimoto, N., Wehrli, C., Wiedensohler, A. and Zhang, X.-Y.: Recommendations for reporting "black carbon" measurements, Atmos. Chem. Phys., 13(16), 8365–8379, doi:10.5194/acp-13-8365-2013, 2013.

Schotterer, U., Stichler, W. and Ginot, P.: The influence of post-depositional effects on ice core studies: Examples from the Alps, Andes, and Altai, in Earth Paleoenvironments: Records Preserved in Mid- and Low-Latitude Glaciers, pp. 39–59, Springer Netherlands., 2001.

Thevenon, F., Anselmetti, F. S., Bernasconi, S. M. and Schwikowski, M.: Mineral dust and elemental black carbon records from an Alpine ice core (Colle Gnifetti glacier) over the last millennium, J. Geophys. Res., 114, doi:D17102 10.1029/2008jd011490, 2009.

Yasunari, T. J., Bonasoni, P., Laj, P., Fujita, K., Vuillermoz, E., Marinoni, A., Cristofanelli, P., Duchi, R., Tartari, G. and Lau, K.-M.: Estimated impact of black carbon deposition during pre-monsoon season from Nepal Climate Observatory-Pyramid data and snow albedo changes over Himalayan glaciers, Atmos. Chem. Phys., 10(14), 13, doi:10.5194/acp-10-6603-2010, 2010.

[Figure]

[Figure]

**Fig. 1.** Annually averaged temporal evolution in rBC mass concentration of the ELB ice cores.